# Street-wise dog testing: Feasibility and reliability of a behavioural test battery for free-ranging dogs in their natural habitat

Svenja Capitain[1,2]*, Giulia Cimarelli[2], Urša Blenkuš[2], Friederike Range[2], Sarah Marshall-Pescini[2]*

1 Department of Physics, Chemistry and Biology (IFM), Linköping University, Linköping, Sweden,
2 Department of Interdisciplinary Life Sciences, Domestication Lab, Konrad Lorenz Institute of Ethology, University of Veterinary Medicine Vienna, Vienna, Austria

* Svenja.capitain@vetmeduni.ac.at (SC); Sarah.marshall@vetmeduni.ac.at (SM-P)

## Abstract

Behavioural scientists are increasingly recognizing the need to conduct experiments in the wild to achieve a comprehensive understanding of their species' behaviour. For domestic dogs (*Canis familiaris*), such progress has been slow. While the life in human households is often regarded as dogs' natural habitat, this classification disregards most of the global dog population. The value of experimentally testing free-ranging dogs' cognition and behaviour is increasingly being recognized, but no comprehensive test batteries have been conducted on those populations so far, leaving the feasibility and reliability of such endeavours unknown. This study is the starting point to fill this gap by pioneering and validating an elaborate behavioural test battery on street-living dogs. Therein, six common temperament tests (human-/conspecific-directed sociability, neophobia, tractability) and dog-human communication paradigms (pointing, inaccessible object) were adapted to the street conditions. We evaluated the feasibility of the test battery, the coding reliability of the measures, and investigated their temporal consistency in a retest of the same individuals six weeks later (test-retest reliability). The test battery proved feasible with most dogs participating in all subtests, and it showed satisfactory inter- and intra-rater reliability (0.84 and 0.93 respectively), providing evidence that complex behavioural tests can be conducted even in highly variable street conditions. Retesting revealed that some behaviours could be captured reliably across time, especially when the subtest was particularly engaging (e.g., human approach, point following). In contrast, the low retest reliability for subtests relying on sustained novelty and behaviours that were highly susceptible to disturbances (e.g., gazing) reflects the difficulties of street dog testing, including standardisation in disturbance-prone environments, ecology-dependent adaptation of methods, and intrinsic differences between pet and free-ranging dogs. With some adaptations, this test battery can be valuable in investigating cognition and behavioural profiles in such an understudied population as free-ranging dogs.

**Data Availability Statement:** All relevant data are within the paper and its Supporting information files.

**Funding:** The research was funded by the Austrian Science Fund (FWF, www.fwf.ac.at: project number P34749 for GC; I5052-B for SMP, UB and FR). The funders had no role in study design, data collection and analysis, decision to publish, or preparation of the manuscript. SC received no specific funding for this work.

**Competing interests:** The authors have declared that no competing interests exist.

# Introduction

The behaviour of dogs has been of central interest to humanity for millennia–possibly since the early stages of dog domestication 15.000 years ago through human-directed selection for specific traits [1–3]. While dogs appear already in the work of early scientists such as Darwin [4] or Pavlov [5], it is only in the last 30 years that the research into dogs' behaviour and cognition has been recognized for its intrinsic value, resulting in an explosion of studies focusing on dogs' cognitive and human-directed capabilities [reviewed by 6,7], their temperament and personality traits [8], and the influence of domestication [reviewed in 9] and our own anthropogenic environment [e.g., as model for aging: 10] on behavioural development.

However, these efforts have primarily revolved around a relatively limited subset of the global dog population–those under immediate human care and control, such as pets, working, and shelter dogs [8]. While easy to access, this subset represents less than a quarter of the global dog population with the other 75–83% consisting of free-ranging dogs [11,12,]. These animals might or might not be owned but share the characteristic that their movement and mate-choice are largely unrestricted by human control [13], even though the majority still resides as street dogs that depend on human settlements for food and shelter [11].

Efforts to explore the behaviour of free-ranging dogs have so far predominantly relied on observational methods [e.g., 14,15, reviewed in 16], leading to important insights into these populations. While such studies are time-intensive and lack the experimental manipulation required to answer some of the questions addressed in the lab, they were so far chosen over experimental approaches due to several challenges that are associated with conducting more standardized test batteries for free-ranging dogs in their natural habitat: Beside the risk of aggression and disease status [e.g., rabies: 17], restraining the animals for a test would be unethical and obstruct the testing of natural behaviour. Hence, tests must rely on voluntary participation and transportable, flexible test setups. Established methodologies and apparatuses thus have to be adapted to the mostly unexplored street conditions, with some inevitable loss of direct comparability, given the different contexts in which animals are tested [18–20]. Finally, this lack of a stable test environment impedes standardisation and the variability and disturbance risk might distort results [21], particularly if the aim is to test the highly dynamic populations repeatedly [reviewed in 16,19,22].

Nonetheless, there are compelling reasons to experimentally assess the behaviour and personality of free-ranging dogs. For one, without studying a representative sample of the global dog population encompassing at least some of the vastly different life experiences and mating choice they may have, our knowledge about dogs' cognitive abilities, human-animal interactions, the connection between genetics and personality, and beyond remains compromised [19, for more specific ideas see 21]. Secondly, it has been shown that, while the admixture of different breeds can be high in free-ranging populations [23], a large proportion is represented by genetically distinct populations that predate the recent strict artificial selection for breed formation [13,23]. Compared to our heavily selected modern breeds where domestication traits might have become uncoupled [24,25], those populations provide important insights for comparison with domestication research both from a genetic and ecological perspective [26]. Finally, free-ranging dogs' pervasive presence in human settlements (e.g., 2,930 dogs/km$^2$ in Kathmandu, Nepal [27]) means that millions of people spend their daily life in dogs' direct proximity. Increasing our understanding of their behaviour and behavioural profiles could therefore improve both human and animal welfare and aid in effective population control [21,26].

Some pioneering groups have already initiated work on free-ranging dogs, demonstrating that an experimental approach, particularly if involving short, one-off tests can indeed be

conducted under the variable yet natural conditions in which they reside [18,20,28–30]. While presenting a valuable start, these tests have focused on single behavioural aspects in short tests [e.g., persistence in 18, pointing in 20], leaving the possibility of longer tests, test batteries, and repeated testing that is needed to explore behavioural profiles, largely unexplored.

Furthermore, critical information regarding the reliability of measures obtained from such tests on the streets remains scarce–information that is vital not only for singular behavioural testing but especially for comprehensive temperament assessments. Several aspects of reliability should therein be considered: the coding reliability, which describes how well the behaviours defined in the ethogram and the testing conditions allow for consistency in a rater's observations over time ("intra-rater reliability") and with others ("inter-rater reliability"), and the "test-retest reliability", which assesses a test's ability to capture the behavioural traits reliably. A reliable test would be expected to elicit the same behaviour in an individual again when being tested in the same test twice ("retest"). This is particularly important in terms of personality traits, which are defined as being consistent over time and context in an individual [8]. Test-retest reliability outcomes are often rather low in pet breed and shelter dogs, but some kind of consistency over time has often been reported [31,32]. Whether this also pertains to free-ranging dogs and/or under highly variable street conditions remains to be explored.

In response to this research gap, our study endeavours to achieve a twofold objective. First, to pilot and evaluate the feasibility of a test battery specifically tailored to street dogs (i.e., free-ranging dogs living in and around human settlements), encompassing a range of behaviours commonly assessed in dog cognition or temperament tests and suggested to be relevant for different domestication hypotheses [tameness: 1, sociability: 2, deferential behaviour: 3, lack of aggression: 33]. These behaviours include human-directed approachability, conspecific sociability using a 'fake dog' test, neophobia and exploration in a novel object test, dogs' understanding of human communication in a pointing test and their use of human-directed communication (gazing) in a begging test, as well as their propensity to shy away from conflict in a tractability test [34–38]. Second, we assess the reliability of this test battery in the natural street setting, including inter-rater, intra-rater and test-retest reliability. We draw conclusions from the results on free-ranging dog testing overall, as well as how this test battery could be adapted to be more widely used in the field.

## Methods

### Ethics statement

Ethical approval for this study was obtained from the Ethical committee at the Agronomic and Veterinary Institute Hassan II (Comité d'Éthique de l'Institut Agronomique et Vétérinaire Hassan II) in Rabat, Morocco (Protocol number: CESASPV_2023_05). The street dogs' participation was voluntary, i.e., dogs were not restrained or forced to take part in the tests, and they were able to leave at any time. Moreover, the procedures were non-invasive and in accordance with the European Union Directive on the protection of animals used for scientific purposes (EU Directive 2010/63/EU).

### Study area and population

The testing of the study subjects was conducted along beach, industrial, and urban areas in the Sous-Massa region in Morocco. The study subjects were part of a big dynamic free-ranging dog population living as scavengers in and around human settlements, hence classifying as street dogs [19]. The behaviour battery was first piloted on a random sample of eight adult street dogs (4 males, 4 females). Since the test battery was finalized after the first three and thus remained the same for the other five pilot dogs, they were added to the final test sample.

Hence, a total of 36 adult street dogs (20 males, 16 females; 5 pilot plus 31 test sample dogs) were tested to assess the feasibility and reliability of the finalized test. Twenty-six of those dogs (15 males, 11 females) were found again for the retest.

## Procedure

The test battery consisted of six short behavioural subtests and three physiological samplings conducted in direct succession (Fig 1). The subtests were designed to resemble behaviour tests that have commonly been used to assess pet, shelter, or working dog temperament traits, dogs' cognitive abilities, or traits that have been suggested as central during the domestication process (see description of the respective subtests). The three physiological samples (saliva and hair) were part of the test battery to assess the feasibility of sampling in such a setting and allow for later use of those samples in future exploration.

For the test-retest reliability measure, the test battery was conducted a second time ("retest") with the same study subjects within an average time span of 46.28 days (min. 33, max. 76 days) which was similar to other test validation procedures [e.g., 39–42]. To this end, the location of the initial test was noted, and the area was frequently visited again after five to six weeks until the dog was found, identified with previous pictures, and tested again. It was noted if the dog was retested in a different location than the first test (i.e., not in visual vicinity of the previous location). The procedure in the first and retest was exactly the same as described below. For the object-based subtests (Fake dog and Novel object subtest, see below), two slightly different-looking versions were used respectively for the test and the retest to create novelty. The order of presentation of the two stimuli was randomized across subjects.

**Initiation and general circumstances.** Two experimenters were involved in each testing occasion: E1 was always the same person (SC) and conducted the actual tests while E2 (three different people) video-recorded, helped set up the subtests, or distracted other dogs if necessary. Dogs were tested individually and had to be alone before and during the test. Tests were conducted between sunrise and 10 a.m. before the areas became too crowded. The experimenters accessed the study area by car and all dogs encountered alone were tested. If at any point during testing, the dog ran off more than 10 m, the subtest during which the dog left was terminated and E1 tried to lure the dog back. If the dog ran too far to be retrieved, or did not respond to the lure, the whole test was terminated. If anyone other than the experimenter approached and visibly distracted the dog (e.g., head turn), the test was interrupted until the criterium was restored and the test resumed. If at any point dogs showed active signs of aggression, the test was terminated (however, this never occurred). For each dog, location, time, weather, health status, and activity upon arrival of E1 (e.g., sleeping, walking, . . .) were noted.

The exact setup of each subtest is displayed in Fig 2, the procedure of each subtest is demonstrated in the supporting information (S1 Video).

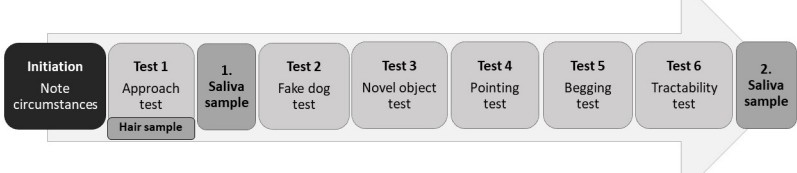

**Fig 1. The sequence of the employed test battery.** Subtests in light and physiological sample collections in dark grey.

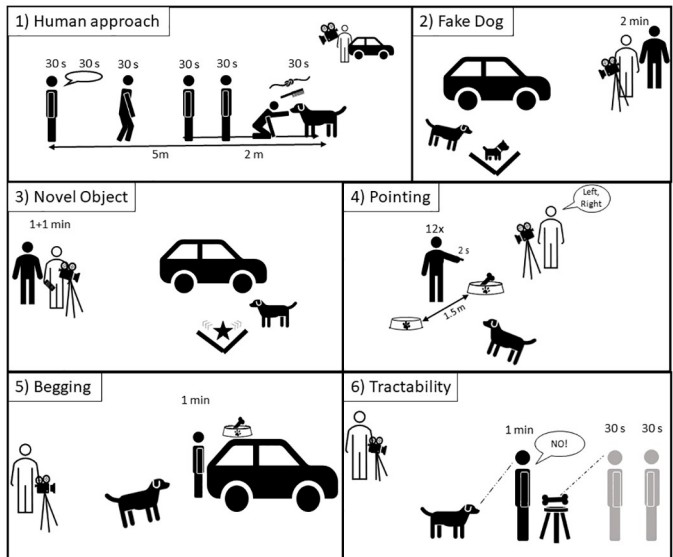

**Fig 2. The set-up of each of the six subtests in the presented test battery.** The main experimenter (E1) is presented in black (or grey in later stages of the subtest), the assistant experimenter (E2) in white.

**Human approach test (subtest 1).** Target: Approachability, human-directed sociability/docility, tameness, play. This test was modelled after similar tests used in widescale personality assessment tests (e.g., the Dog Mentality Assessment (DMA) in the Swedish Kennel Club [43] and has in part been used with Ethiopian village dogs [30]. Beside the assumption that human-directed sociability is an essential trait under selection during the domestication process [2], assessing the actual human-directed behaviour in a free-ranging population is provides valuable insights for disease- and population control.

Here, E1 stepped out of the car and whistled shortly to grab the dog's attention. She then stepwise proceeded to 1) stand quietly looking at the subject for 30 s, 2) called the dog whilst making friendly gestures (snapping, tapping on leg) for another 30 s, 3) approached the dog for 30 s up to 2 m distance, 4) stood for another 30 s, 5) approached the dog to one body length distance and paused another 30 s, and 6) crouched for another 30 s all while continuing to call the dog over. If the dog approached to less than 10 cm of E1 during any of those phases, E1 kneeled and tried to pet the dog for 30 s. If tolerated, E1 used a small brush to collect hair samples. The dog was then invited to play tug of war with a rag toy (1 m long) for 30 s.

**Fake dog test (subtest 2).** Target: Conspecific-directed sociability and aggression. Several studies support the idea that stuffed dogs can be used to reliably assess dogs' initial behaviour towards conspecifics [34,44]. A reduction in conspecific aggression has been hypothesised as an important shift from wild to domestic types [45], and understanding intraspecific aggression in free-ranging dogs can provide important applications for the local population.

Hence, subjects were exposed to a "Giant Jack Russel Terrier" stuffed fake dog (brand: 'Melissa & Douk'; wither height 28 cm; S2 File). The two used versions differed in ear posture (pointy or droopy), tail length (medium and long), and colour (one original, one wrapped in dark bags) in the test and in the re-test respectively, in a randomized order across subjects. The fake dog was positioned 5 m from the car behind a V-shape visual barrier with its opening towards the car. This visual barrier allowed only the approaching subject but no other surrounding dogs to see the fake dog. E1 guided the subject towards the fake dog (if needed with

food) and quickly walked >5 m away as soon as the subject saw the fake dog. The subject's behaviour was observed for 2 min.

**Novel object test (subtest 3).** Target: Neophobia, exploration/novelty-seeking. Presenting a dog with an unknow object is commonly used to assess novelty-seeking and boldness [38,46], and explorative-behaviour is hypothesised to be lower in domesticated animals as part of the domestication syndrome [45].

To present an object that was truly novel to the dogs, we fixed a 90–100 cm high foil balloon with a wooden stick on top of a remotely controlled car (31x18x18 cm, Model: DEERC DE42 RC, S2 File). Two balloon forms were used (dinosaur, a number (zero, five) sprayed in neon green and blue) for test and retest and randomized between study subjects. The set up was the same as the Fake dog test. As soon as the dog saw the novel object, E2 started moving it back and forth for 1 min through remote control. The toy was then left inactive for another 1 min. The dog's behaviour was observed for the full 2 min.

**Pointing test (subtest 4).** Target: Dog-human communication skill, willingness to follow commands, cooperative communication [reviewed in 47,48]. Studies have shown that a wide array of species, including canids, can follow pointing cues [for review see 47]. Exploring this phenomenon in free-ranging dogs with differing levels of human experience will further our understanding of the role of life-experience vs. species predisposition (but see [20] for pioneering work in this regard).

The test started with four warm-up trials: E1 placed a baited bowl (12 cm diameter, 10 cm height) on the ground 2 m from the dog, standing 1 m away herself and verbally called the dogs attention. After the dog ate the food, the bowl was picked up, baited, and placed down again to familiarize the dog with the fact that the bowl contained retrievable food. The subtest was terminated if the dog did not approach the bowl within 30 s two times in a row. The warm-up trials were followed by twelve test trials: one empty and one baited bowl (both rubbed in food for smell, baited outside of the dog's view) were placed 1.5 m apart 2 m from the dog. E1 stood equidistant behind the bowls, attracted the dog's attention, and upon eye contact, pointed to the baited bowl for 2 s (momentary distal pointing). The dog was allowed to eat the food if it approached the correct bowl on the first choice, otherwise the bowls were immediately collected by E1. Each trial could be repeated twice if the dog did not choose within 30 s, followed by one repetition of the warm-up trial (i.e., only one bowl with food). If after the third repetition, the dog did not choose, the subtest was terminated. We used six different randomized sequences related to the positioning of the baited/pointed bowl to the right or left of the experimenter (e.g., RLRLLRLRRLLR) that were called out to E1 by E2.

**Begging test (subtest 5).** Target: human-directed communication, gazing behaviour, suggested to be help-seeking behaviour [35, reviewed by 36, but see for alternative explanations: 49,50]. Also a behaviour suggested as a trait under selection during domestication but the wide variability between individuals depending on human socialisation calls for further investigation with differentially socialised populations [51,52].

The dog was allowed to eat from the baited bowl three times. Afterwards, the bowl was visibly baited and placed out of reach but still visible on the car, a wall, or a tree. E1 stepped 1 m to the side and faced away from the bowl with a neutral facial expression. The dog's behaviour was observed for 1 min.

**Tractability test (subtest 6).** Target: Social inhibition, tractability/deferential behaviour (i.e., the ease to yield to higher command, conflict avoidance [3]). According to the deferential hypothesis [3], tractability has been a central selection factor in dog domestication. Furthermore, assessing dogs' inclination to avoid conflicts with humans in a resource scenario may also have it uses in a more applied setting.

E1 gave the dog a piece of food, then took a big piece of meat, showed it to the dog, and visibly placed it on top of a small stool (40 cm height). E1 loudly stepped between dog and stool, stared the dog in the eyes and firmly said "No, it's mine!". If the dog attempted to get to the stool, E1 loudly stepped in their way and repeated "No" in a firm tone. After 1 min of this, E1 stepped 1 m to the side and stood still for 30 s while looking at the food. Finally, E1 turned around and looked away for 30 s. The test was finished as soon as the dog ate the food or after the last 30 s.

## Video coding

The tests were recorded by an action camera (4K 24fps WiFi Sports Action Camera or Action Cam Jeemak WiFi) fixed on E1's chest and a camera (Sony HDR-CX405 Full HD Camcorder, hand-held (subtest 1) or on a tripod (subtest 2–6)) handled by E2. BORIS (version 7.13.8, (Friard & Gamba, 2016)) was used for the video coding.

The ethogram aimed at capturing the target behaviours described for each subtest. We used a bottom-up approach by initially coding 55 variables with different modifiers (i.e., the person or object the behaviour was coded towards) that measured almost every potentially relevant aspect in the different subtests. The coded variables are displayed in Table 1 with a detailed ethogram in S1 Table. Social behaviours were based on the ethograms used at the Wolf Science Center of the University for Veterinary Medicine, Vienna [9], and to observe free-ranging dogs in Italy [15,53].

This list of variables was narrowed down by 1) assessing their codability through the inter- and intra-rater analysis (for acceptable agreement see Statistical Analysis), 2) assessing the variables' centrality in the subtest by analysing their frequency (i.e., excluding variables that were displayed in less than 10% of tests), and 3) assessing their consistency through the test-retest reliability. Only behavioural variables that passed step 1) and 2) went on to be analysed in step 3) to ensure that the test-retest reliability was based on reliably coded and not overly skewed data [54].

For the intra-rater reliability measures, a random subsample of 20% of the test videos (n = 13; balanced between test and retest, full test batteries and tests that were terminated halfway through) were re-coded by the initial rater after three weeks. For the inter-rater reliability, a random subsample of 20% of the full-test videos (n = 9) were coded independently by a second rater [for similar methods see 38,55]. Beforehand, the second rater was trained on three sample videos and the ethogram was further clarified where needed.

## Statistical analysis

Inter- and intra-rater reliability was analysed for each behavioural variable summed across all subtests and modifiers to assess their codability. Test-retest reliability, on the other hand, was analysed for every single variable by modifier and subtest. Occurrence variables (e.g., in which phase the dog approached, termination, etc.) were converted into ordinal variables before analysis. To normalize the data for comparison between test and retest, the time the dog was not visible or a disturbance was coded was subtracted from the total time of the subtest of the individual, and the behavioural data was then divided by this normative time. All individual subtests in which more than 70% of the time was coded as disturbance and/or not visible were excluded from further analysis. In case of the test-retest reliability, the specific subtest in both the test and the retest were excluded to allow for fair comparison.

The statistical analysis was then conducted in R (Version 4.2.2 [56]). To account for the systematic errors between raters/rating occasions, intraclass correlation coefficients (ICCs) were calculated to assess the reliability measures [57,58]. Additionally, considering that both raters/

**Table 1. Coded behavioural variables and occurrence across subtests.**

| Category | Behavioural variable | Human approach | Fake Dog | Novel Object | Pointing | Begging | Tractability |
|---|---|---|---|---|---|---|---|
| **Proximity** | Close | E1 | Fake dog, E1+E2 | Novel object, E1+E2 | - | E1 | E1, stool |
| | Medium | E1 | Fake dog, E1+E2 | Novel object, E1+E2 | - | E1 | E1, stool |
| **Tail** | Wagging | E1 | Fake dog | Novel object | E1 | E1 | E1 |
| | Between legs | Yes | Yes | Yes | - | Yes | Yes |
| **Gazing** | | E1 | Fake dog | Novel object, E1+E2 | - | E1 | E1 |
| **Vocalization** | Barking | E1 | Fake dog | Novel object | - | E1 | E1 |
| | Growling | E1 | Fake dog | Novel object | - | E1 | E1 |
| | Whining | E1 | Fake dog | Novel object | - | E1 | E1 |
| **Displacement** | Nose-licking | Yes | Yes | Yes | - | Yes | Yes |
| | Body shake | Yes | Yes | Yes | - | Yes | Yes |
| | Stretching | Yes | Yes | Yes | - | Yes | Yes |
| | Yawning | Yes | Yes | Yes | - | Yes | Yes |
| | Sniffing the ground | Yes | - | - | - | - | - |
| **Physical contact** | Biting | E1 | Fake dog | Novel object | - | E1 | E1 |
| | Body contact | E1 | Fake dog | Novel object | - | E1 | E1 |
| | Jumping | E1 | Fake dog | Novel object | - | E1 | E1 |
| | Licking | E1 | Fake dog | Novel object | - | E1 | E1 |
| | Mouthing | E1 | Fake dog | Novel object | - | E1 | E1 |
| | Pawing | E1 | Fake dog | Novel object | - | E1 | E1 |
| | Sniffing | E1 | Fake dog | Novel object | - | E1 | E1 |
| **Reactions** | Stand tall | E1 | Fake dog | Novel object | - | E1 | E1 |
| | Bare teeth | E1 | Fake dog | Novel object | - | E1 | E1 |
| | Lunge | E1 | Fake dog | Novel object | - | E1 | E1 |
| | Head dip | E1 | Fake dog | Novel object | - | E1 | E1 |
| | Belly exposure | Yes | Yes | Yes | - | Yes | Yes |
| | Flee | E1 | Fake dog | Novel object | - | E1 | E1 |
| | Crouch | E1 | Fake dog | Novel object | - | E1 | E1 |
| | Risk assessment | E1 | Fake dog | Novel object | - | E1 | E1 |
| | Play | E1 | Fake dog | Novel object | - | E1 | E1 |
| | Friendly | E1 | Fake dog | Novel object | - | E1 | E1 |
| | Bow | E1 | Fake dog | Novel object | - | E1 | E1 |
| **Marking** | | Yes | Yes | Yes | - | Yes | Yes |
| **Defecating** | | Yes | Yes | Yes | - | *Yes* | Yes |
| **Non-visible** | | Yes | Yes | Yes | Yes | Yes | Yes |
| **Disturbance** | | Yes | Yes | Yes | Yes | Yes | Yes |
| **Subtest specific variables** | | | Latency to approach fake dog | Latency to approach novel object | Observation of gesture: no/yes | Attempts to reach bowl | Latency to eat food |
| | | Phase of first approach | | Phase of first approach | Successful choice: no/yes | | Phase of eating food |
| | | | Genital sniffing | | No choice | 2-way gaze alternation | |
| | | | | | | 3-way gaze alternation | |

The table shows all behavioural variables (column 2) of a certain category (column 1) that were coded in the respective subtests (columns 3–8). The words in the subtest columns describe towards what person or object the behaviour was coded, while 'yes' means that the behaviour was coded regardless of direction, and a minus (-) indicates that the behaviour was not coded in the subtest. Behavioural variables were either coded as duration (D), frequency (F), occurrence (O), or latency since the beginning of the respective subtest (L). Coloured cells indicate that the variable occurred in less than 10% of tests and was thus excluded from the test-retest analysis.

rating occasions coded the same videos and that the goal was to generalize the reliability results, a two-way random effects ANOVA with an absolute agreement estimate was used to assess the inter- and intra-rater reliability [reviewed by 59]. According to the classification by [60], an ICC below 0.5 is considered poor reliability, between 0.5 and 0.75 it is moderate, between 0.75 and 0.9 it is good, and between 0.9 and 1.0 the reliability is considered excellent. For a behaviour to be regarded as having acceptable coding reliability and move on to the test-retest analysis, the ICC threshold for the inter- and intra-rater reliability was set to a moderate 0.70 [61] which is in accordance with similar studies [weighted average of 0.68 in a review of six studies by 8, 0.77 in 62]. However, careful reconsideration was employed in case the threshold was just about missed by taking both coding reliabilities (inter- and intra-rater) into consideration and examining the p-value (assuming significance level α = 0.05). For the test-retest reliability in the repeated sample, a two-way mixed effects ANOVA with absolute agreement estimate was applied [60]. Since significance largely depends on sample size and significant test-retest reliabilities in pet dog tests commonly range around an ICC of 0.5 which would usually be classified as poor to moderate reliability [average ICC of 0.43 in a meta-analysis of 31 studies in 31, e.g., 0.58 in 32, 0.57 in 48], a test-retest reliability with a p-value below 0.05 rather than a specific ICC was chosen as reliability indicator. Only variables that were displayed in more than 10% of the testing occasions were included in the test-retest reliability to circumvent unreliable analysis of heavily zero-inflated and low-prevalence data [for a similar approach see 54,61,63]. For the ordinal variables, two-way ANOVA's with absolute agreement estimate (equivalent to weighted Cohen's Kappa according to [64]) were used to analyse the coding and test-retest reliability. If the ICC was poor (i.e., below 0.5) for a certain variable, systematic biases in the measurements of both rating occasions were explored through scatter plots and variables were excluded if necessary. To control for the test-retest reliability being low due to disturbances during the tests, we conducted the same analysis again but excluded all individual subtests in which any disturbances were coded (46 of 235 subtests). Disturbance had been coded when the dog clearly turned its head towards an interfering human/dog/other animal for more than three seconds during a running subtest and while the test subject was visible on the video.

Lastly, because curiosity- or exploration-based tests may be more prone to lower consistency due to repeated exposure to similar stimuli [46], we tested if the dogs systematically showed a decreased reaction (exploration and fear) in the fake dog and novel object retest compared to the first test. To that end, applicable variables that were retained in the test-retest reliability (i.e., had good coding reliability and occurred in more than 10% of tests) were summed into an exploration-related and a fear-related variable [therein drawing from 65,66] for each of the two subtests respectively. After transforming these four left-skewed variables with a square root transformation, a paired t-test was used to compare whether they differed significantly between test and retest.

## Results

### Subject characteristics

Out of the 36 initially tested street dogs, 23 (14 males, 9 females) were tested in the entire test battery, while nine dogs (5 males, 4 females) terminated the test during the Human approach subtest and four dogs (1 male, 3 female) after the first two or three subtests by running away. For the retest, 26 and thus 72% of the 36 initially tested street dogs were found again: 19 of those were dogs who went through the full test battery initially and 7 were dogs who had run away during the first testing. Fourteen were retested in a different location than the first test. All 26 dogs completed the same number of subtests in the retest as they did in the initial test.

## Changes in methods and variables

After the test was adapted in the pilot testing, some additional changes from the above-presented procedure had to be made throughout the main testing period. For one, the play-part (rope presentation for tug-of-war) in the Human approach test caused four (28.57%) of the first 14 dogs to run away and terminate the test. For two more, the play phase had to be terminated to keep the dog from leaving. Thus, the play phase was dropped half-way through the main sample and the variable 'Interacting with the toy' was excluded from the analysis.

In addition, two more variables had to be excluded due to the following reasons: 'Auto-grooming' because the dogs had varying amounts of parasites, wounds, and dirt in their fur which influenced the scratching rate, and 'Sniffing the ground' in all but the Human approach test because food was dropped on the ground during the subtests and the different ground surfaces influenced the dogs' sniffing behaviour.

## Intra- and inter-rater reliability

For the intra- and inter-rater reliability coding, almost all variables passed the target ICC-threshold of 0.7 and had a significant p-value at 0.015 or below (the exact statistical outputs for each variable are reported in S2 Table). The average of both inter- and intra-rater reliability for all variables was above the target threshold of an ICC of 0.7, with a good reliability of 0.84 in the inter-rater and 0.93 in the intra-rater analysis.

There were only few variables that did not clear the threshold. 'Tail between the legs' and 'Mouthing' had moderate inter-rater reliability, meaning they still had an ICC above 0.5, and the p-value was below 0.05. Likewise, 'Medium proximity' had a moderate reliability within and between raters. The inter-rater reliability for 'Head dip', 'Friendly' and 'Stand tall', on the other hand, were poor with an ICC below 0.5 with a p-value above 0.05.

While 'Tail between the legs' and 'Mouthing' did not clear the initially defined threshold, they were both close and had a p-value of 0.02 which is substantially below the usual 0.05 significance level. Since previous studies have deterred from the arbitrary 0.7 ICC threshold in favour of significant p-values (p<0.05) [8], and both present crucial behaviours in dogs, we tentatively chose to keep them in the analysis going forward but treat their interpretation with utmost care. The other behaviours fell below the 0.05 significance level and were thus excluded from further analysis.

## Exclusion of infrequent variables

In the next step, variables that did not occur in more than 10% of the respective subtests were identified (Table 1) and excluded from the test-retest reliability analysis. In short, close proximity, tail wagging, gazing, nose licking and yawning were the only variables that occurred frequently in all subtests. On the other hand, growling, whining, stretching, shaking, biting, licking, mouthing, marking, defecating, baring teeth, lunging, belly exposure, and bow were not displayed in more than 10% of tests in any of the subtests. Even after summing most of them into one pooled 'stress' variable [following the classification of 67,68], they did not pass the 10% occurrence threshold and were hence excluded from the test-retest analysis. All subtest-specific variables except 'attempt to reach the bowl' in the begging test, occurred frequently enough to pass the 10% threshold.

## Test-retest reliability

The final test-retest reliability analysis was conducted for each variable by subtest and modifier, only taking the variables into account that 1) were not already dropped from the

ethogram/tests throughout the testing (i.e., auto-grooming, sniffing the ground, play), 2) had an acceptable coding reliability, and 3) occurred in more than 10% of all tests. The resulting test-retest reliabilities are reported in Table 2.

The average test-retest reliability was poor with an average ICC of 0.23. To see whether the poor outcome was influenced by disturbances during the tests that altered the dogs' behaviour, we conducted the same analysis but excluded all subtests in which a disturbance had been coded (i.e., when the dog clearly turned its head towards an interfering human/dog/other animal for more than three seconds during a running subtest and while the test subject was visible on the video). While this changed the reliability of some behaviours, the average test-retest reliability did not improve (Table 2; the exact statistical outputs of both analyses can be found in S3 Table).

To summarize by subtest, the Human approach test performed well with ten out of 15 behaviours showing significant test-retest reliability, resulting in an average test-retest reliability of 0.32. In the Fake dog test, only 'tail between the legs' had a moderate retest reliability, with 'close proximity to the fake dog' and 'nose-mouth licking' also being moderately reliable after tests with disturbances were removed. The average test-retest reliability was low (ICC = 0.06). Similarly, only 'close proximity to E1 and E2' were significant in the Novel object test, being joined by 'nose-mouth licking' after disturbed tests were excluded, resulting in an average test-retest reliability of 0.12. The average reliability of the Pointing test was significant with an average ICC of 0.51 with all subtest-specific variables, as well as 'tail-wagging', showing moderate test-retest reliability. The Begging test had a low test-retest reliability (average ICC = 0.16) with only the 'close proximity to E1' being significant across time when only looking at tests without disturbances. Lastly, 'close proximity to E1', 'gazing at E1', the 'phase in which the dog ate the food', 'three-way gaze alternation between person and stool' and 'fleeing' had significant reliability in the Tractability test, with the addition of 'nose-mouth licking' after disturbances were removed. The average test-retest reliability was 0.22.

## Test-retest comparison for the curiosity-based subtests

The summed variables constituted of the following behaviours: "Exploration"-variable: 'close' and 'medium proximity', 'gazing', 'tail wagging', 'friendly', and 'sniffing' the fake dog/novel object, as well as 'genital sniffing' in the case of the fake dog. "Fear"-variable: 'head dip' and 'tail between the legs' for the fake dog subtest. Only 'flee' was retained as fear-representing behaviour in the novel object but was displayed so rarely, that it could not be analysed here.

The dogs showed significantly less exploration and less fear behaviours towards the fake dog in the retest than the first test (exploration: t(14) = 3.16, p<0.01; fear: t(15) = -3.08, p<0.01). Exploration behaviours shown towards the novel object did not differ between test and retest (t(11) = 1.0, p = 0.34).

## Discussion

In this study, we presented what, to our knowledge, is the first attempt at testing free-ranging dogs in a behavioural test battery and report feasibility and reliability of the test measures. Testing the dogs in their natural environment was indeed feasible and the coding reliability was high, indicating that this (relative to previous studies) long and complex behaviour test battery can be conducted with street dogs. The range from good to poor test-retest reliability, on the other hand, reflects the challenges inherent in conducting behaviour tests with free-ranging dogs on the streets.

Although the subtests in this test battery were adapted from mostly 'lab-based' behaviour assessments [e.g., 34,38,46] [but see 20], standardizing the set up and testing our dog

**Table 2. Summary of the test-retest reliability in all retained variables.**

| Category | Behavioural variable | Human approach | | | Fake Dog | | | Novel Object | | | Pointing | | | Begging | | | Tractability | | |
|---|---|---|---|---|---|---|---|---|---|---|---|---|---|---|---|---|---|---|---|
| | | M. | All | w/D | M. | All | w/D | M. | All | w/D | M. | All | w/D | M. | All | w/D | M. | All | w/D |
| **Proximity** | Close | E1 | 0.71 | 0.77 | FD | 0.20 | 0.74 | NO | -0.09 | 0.06 | - | | | E1 | 0.8 | 0.76 | E1 | 0.75 | 0.72 |
| | | | | | E1+E2 | -0.08 | -0.17 | E1+E2 | 0.74 | 0 | | | | | | | Stool | -0.04 | -0.02 |
| **Tail** | Wagging | E1 | 0.61 | 0.62 | FD | 0.14 | -0.12 | NO | 0.04 | NA | E1 | 0.5 | 0.2 | E1 | 0.36 | 0.3 | E1 | -0.14 | -0.06 |
| | Between legs | Yes | 0.53 | 0.52 | Yes | 0.71 | 0.72 | | | | | | | | | | | 0.29 | 0.28 |
| **Gazing** | | E1 | **0.35** | 0.34 | FD | 0.16 | 0.11 | NO | -0.35 | -0.06 | | | | E1 | 0 | 0.24 | E1 | **0.48** | 0.44 |
| | | | | | | | | E1+E2 | -0.26 | -0.08 | | | | Bowl | 0.34 | 0.12 | Stool | 0.18 | 0.14 |
| **Vocalization** | Barking | E1 | 0.85 | 0.85 | FD | 0.12 | 0.12 | | | | | | | | | | | | |
| **Displacement** | Nose-licking | Yes | 0 | -0.01 | Yes | 0.16 | 0.56 | Yes | 0.37 | 0.61 | - | | | Yes | -0.2 | -0.28 | Yes | 0.41 | 0.51 |
| | Yawning | Yes | -0.03 | -0.08 | Yes | -0.06 | 0 | Yes | 0.48 | 0.45 | - | | | Yes | -0.09 | -0.1 | Yes | -0.21 | -0.2 |
| | sniffing the ground | Yes | -0.04 | -0.04 | Excl. | | | Excl. | | | - | | | Excl. | | | Excl. | | |
| **Physical contact** | Body contact | E1 | 0.55 | 0.55 | | | | | | | | | | | | | | | |
| | Jumping | E1 | 0.5 | **0.46** | | | | | | | | | | | | | | | |
| | Sniffing | E1 | 0.1 | 0.05 | FD | 0.03 | 0.19 | NO | -0.04 | 0 | - | | | E1 | -0.09 | -0.05 | | | |
| **Reactions** | Flee | E1 | -0.08 | -0.16 | | | | NO | 0.32 | NA | | | | | | | E1 | 0.76 | 0.58 |
| | Crouch | | | | | | | | | | | | | | | | E1 | -0.05 | -0.02 |
| | Friendly | E1 | **0.45** | 0.56 | FD | -0.07 | 0.15 | | | | | | | E1 | 0.34 | 0.35 | E1 | 0.13 | 0.14 |
| | Head dip | E1 | -0.08 | -0.09 | FD | -0.21 | 0 | NO | 0 | NA | | | | | | | E1 | -0.07 | -0.02 |
| | Play | E1 | -0.03 | -0.06 | | | | | | | | | | | | | | | |
| | Risk assesment | | | | FD | -0.04 | -0.04 | | | | | | | | | | | | |
| | Stand tall | | | | FD | -0.12 | -0.29 | | | | | | | | | | | | |
| **Subtest-specific variables** | Phase of first Approach | E1 | **0.49** | 0.51 | | | | NO | 0.2 | 0 | - | | | - | | | Stool | 0.65 | 0.75 |
| | Latency of first approach | | | | FD | -0.15 | 0.2 | NO | 0.05 | -0.9 | - | | | - | | | Stool | -0.11 | -0.08 |
| | Genital sniffing | | | | FD | 0.15 | -0.15 | | | | | | | | | | | | |
| | Observation of gesture: yes | - | | | - | | | - | | | E1 | 0.51 | 0.12 | - | | | - | | |
| | Observation of gesture: no | - | | | - | | | - | | | E1 | **0.33** | -0.2 | - | | | - | | |
| | Successful choice | - | | | - | | | - | | | Bowl | 0.64 | 0.38 | - | | | - | | |
| | Unsuccessful choice | - | | | - | | | - | | | Bowl | 0.55 | 0.06 | - | | | - | | |
| | No choice | - | | | - | | | - | | | Bowl | 0.52 | 0.17 | - | | | - | | |
| **Gaze alternations** | 2-way | - | | | - | | | - | | | - | | | E1/bowl | 0.03 | -0.08 | E1/stool | 0.14 | 0.07 |
| | 3-way | - | | | - | | | - | | | - | | | E1/bowl | 0.1 | 0.08 | E1/stool | **0.38** | **0.38** |
| **Average per subtest** | | | **0.32** | **0.32** | | **0.06** | **0.13** | | **0.12** | **0.10** | | **0.51** | **0.12** | | **0.16** | **0.13** | | **0.22** | **0.23** |
| **Average across the test battery** | | | **0.23** | **0.17** | | | | | | | | | | | | | | | |

For each variable and subtest, the modifier (M.) and the test-retest reliability is displayed for all tests (All) and for only the tests did not have any coded disturbances (w/D) respectively. The modifiers were the main experimenter (E1), the fake dog (FD), the novel object (NO), the bowl, or the stool. Test-retest reliabilities with a p-value below 0.05 were regarded as acceptable. Behaviours with non-significant test-retest reliability are indicated with a light-grey background and in bold if they were significant but regarded as poor reliability (i.e., below an ICC of 0.5). Variables that were excluded due to low occurrence (i.e., less than 10% of testing occasions) are not shown unless their exclusion only applied to certain subtests. In that case, they are left empty or marked with NA if it only applies to the analysis without disturbances. Variables marked with a minus (-) were not coded in the respective subtest. Detailed analysis results with the 95% confidence interval, F-value statistic, and p-value can be found in S3 Table.

population in the streets turned out to be surprisingly manageable. Importantly, there were no terminations due to aggression towards people or equipment and most dogs completed all six subtests and sample collections despite an average test duration of 30 minutes, suggesting that street dogs (at least those of this population) can remain motivated throughout longer and more complex procedures than those used so far [e.g., 18,29,30,69].

Similarly, the concern that testing and filming in the variable street conditions would negatively impact the coding reliability was unfounded. Our results surpassed the target of a mean ICC of 0.7 with good average inter- and intra-rater reliability of 0.84 and 0.93 respectively. Contrary to our expectations, the coding reliability in this street test was thereby *on par* or even better than many pet and working dog studies [ICC 0.68 in meta-analysis by 8, ICC >0.9 in 38, ICC 0.77 in 62], suggesting that our chosen variables were not only well defined but could also be coded reliably despite the more challenging conditions. Similarly encouraging inter-rater results have been reported for a few other street dog tests as well [18,20], further highlighting that such results are achievable for street testing in general.

However, when looking at the reliability of the individual variables, a few did not meet the threshold. Notably, 'medium proximity', i.e., two body lengths distance from the experimenter, was the only variable which was low both in the inter- and intra-rater reliability analyses. With the constant adaptation of the camera angles to adjust for the variability of the street conditions, it might thus be best to stick to distances that are easier to identify such as the more reliable 'close proximity' (i.e., one body length distance). Additionally, 'head dip', 'stand tall', and 'friendly' all had poor agreement between the raters, which might be due to the complex definitions of these behaviours (S1 Table). While these definitions were adapted from previously used ethograms [16,70,71], the filming in street conditions likely made the subtle cues harder to detect. Given the poor coding reliability of those traits, they should thus be dropped from the presented ethogram unless a more reliable definition or training method can be found and validated, and care should be taken in the establishment of future tests to balance between the complexity of target behaviours and filming feasibility in the streets.

While these outcomes paint a positive picture for street dog testing so far, the feasibility in terms of capturing the desired traits cannot be claimed without looking at the test's ability to elicit the same behaviours consistently over time. The presented test battery resulted in an average test-retest reliability of 0.23, which, according to [60], is regarded as poor reliability. While the definition of personality traits is firmly bound to their consistency over time and across contexts [8], capturing such behavioural consistency has been an issue for many test batteries. A meta-analysis of 31 studies revealed an average trait consistency of 0.43 in pet, shelter and working dogs, which was considered significant although the studies within displayed highly variable results ranging from -0.73 to 1.00 [31]. We found a similarly high variability in the test-retest reliability of the behaviours we assessed, with subtest reliability varying between a moderate ICC of 0.51 for the pointing test and a poorly reliable 0.06 for the fake dog test. Capturing consistent personality or behavioural scores over time is hence an issue not unique to the testing of street dogs in their natural habitat.

Intriguingly, most of the variables that were reliable over time were human-directed behaviours, including proximity to and tail wagging at the experimenter, body contact, and time of first approach. Likewise, the subtests in which the experimenter constantly interacted with the dog (i.e., human approach, pointing, tractability test) had the highest average test-retest reliability over time. Our results are consistent with other studies that report high reliability and strong heritability for sociability and biddability in pet dogs [e.g., 38,46,72] as well as dogs' performance during pointing tasks [73,74], which might tentatively point to a genetic basis for these traits [2]. Simultaneously, while it has been shown that dogs already follow such pointing cues as pups [75] and that their performance is stable regardless of keeping condition, time

spent with the owner, or training [73], we cannot rule out that the individuals' life experience with human food provisioning played a part in these stable outcomes (as human socialisation seems to play a big part in pointing performance in other species [e.g., wolves: 75, goats: 76]).

More important in terms of street dog testing, however, is that the discrepancy in reliability between these social tests and the other subtests (fake dog, novel object, and begging test in particular) might be driven by the former's high interactivity, leaving the animal little time to be distracted. Contrary to pet dogs, free-ranging dogs have been shown to gaze at an inattentive person less than at an attentive person [29]. Moreover, testing in the free-ranging dogs' natural habitat means testing in highly variable and frequently disturbed environments. We tried to account for this by excluding all subtests with coded disturbances in a second analysis, however this did not improve reliability, possibly because we underrepresented the disturbances to our study subjects. Indeed, only six of the 35 dogs were alone in both tests (i.e., initial test and retest). For all others, additional dogs appeared during the test(s) and had to be 'distracted away' by the second experimenter. The appearance of other dogs is an unavoidable occurrence, and whilst testing can continue in such cases, it might explain why the subtests in which the dog had more time to focus on their environment without the direct engagement by the experimenter (fake dog, novel object, begging) showed a lower retest reliability. Whether behaviours like gazing and yawning and the entire begging test (which does not only lack interaction but also novelty) can reliably reflect a dog's behaviour under such circumstances is hence questionable and requires further rigorous testing in alternative test setups on the streets.

To address the disturbance problem on the streets, we propose four not mutually exclusive approaches: 1) Have at least one additional person on the team that is solely responsible for distracting approaching dogs out of sight. 2) Allow for longer between-subtest-phases if the disturbance is temporary and let the dog return to an attentive state rather than pushing through the test. However, one should keep in mind that this changes the time dogs spend in the overall test procedure and carefully consider an influence on the respective goal. 3) If the distraction is substantial, terminating the test and either repeating (though the measures from the initial testing should be used for the repeated subtests) or resuming the test on a different day might be an option, depending on the respective test set up, goal, and familiarity with the population. 4) Meticulously coding any distraction might allow for correction of the disturbances, or at least provide deeper understanding of their influence on the behaviour, based on which decisions for future testing can be made.

Even after considering the disturbance problem, the consistency of the dogs' behaviour in the fake dog and novel object tests was notably low. This might seem surprising since both tests are regularly used in dog temperament assessments [e.g., 34,63]. However, we could not find any reference to the retest reliability of fake dog tests. Moreover, even highly standardised studies exploring novel object tests and novelty-seeking and fearlessness traits often showed poor test-retest reliability (i.e., ICC<0.50) [e.g., Novelty-seeking: 0.48 in 38, Environmental Sureness: 0.16 in 62]. Accordingly, it has been proposed that test-retest outcomes are sensitive to the novelty of the object and situation due to learning effects [46]. Indeed, the fact that the free-ranging dogs explored the fake dog significantly more in the initial test while showing less fear in the retest supports the assumption that the animals partially remembered the initial encounter and adapted their behaviour accordingly. Although we made efforts to modify the appearance of the fake dog and novel object for the retest, the situation, procedure, and set-up remained the same and seem to have lacked the novelty needed to "fool" the animals twice within just six weeks. Nevertheless, it needs to be stressed that the street dogs frequently displayed appropriate conspecific, startling, and investigation behaviour in the initial presentation. Additionally considering that those subtests have been validated with fake vs. real dogs in

case of the fake dog test [34,44] and deemed reliable despite similar retest problems in case of the novel object [e.g., 38,46], we suggest that these two subtests are still suitable to test the street dogs when only a single exposure to the test is needed. Additionally for the fake dog test, the social behaviours towards the stimulus were mostly exhibited in the first half of the subtest, suggesting that it is the initial reaction to the fake dog that is the most representative of conspecific behaviours. A shorter test duration and/or movement in the fake dog [e.g., 44] could thus be considered to decrease not only the influence of distractions, but also the risk that neophobia and boldness traits replace the conspecific behaviours as soon as the dogs realize that the fake dog is not in fact real.

At the same time, these two subtests act as a reminder of the importance of appropriate between-test-intervals, particularly for test batteries investigating behavioural profiles. With free-ranging dogs, the balance between waiting out potential learning effects and retesting early enough to find a considerable number of test subjects again can be difficult, especially if the study population is not well-known. Our substantial retesting rate of 72% within an average of six weeks was hence surprising, though the considerable variation between the individuals (min. 33, max. 76 days) has to be acknowledged. Meticulous notes on appearance, location, and particular behaviours were crucial in this effort. While working with a well-known population would streamline such efforts, our results show that it is not a prerequisite. In addition, using fewer memorable cues (e.g., the V-shaped wooden board in the Fake dog and Novel object subtest) might reduce learning effects and allow for a shorter between-test-interval with better outcomes than presented here.

Beside these two probable explanations for the low test-retest reliability in some subtests, an additional–though highly speculative–explanation might be that free-ranging dogs could be less genetically fixed in some behavioural traits than modern breed dogs and thus behave less consistently in general. This idea might not be too farfetched, considering that free-ranging dogs are still subject to free mate choice and the instability of their environment [13]. One might therefore speculate that behavioural plasticity for certain traits (e.g., adaptiveness to the highly dynamic group compositions, weather conditions, or anthropogenic changes) might be a valuable and thus widely retained trait in free-ranging dogs to deal with those constant environmental changes. Meanwhile, the directional selection for certain reliable breed characteristics in modern breed dogs [77,78] and the resulting considerable loss of genetic variation [79] might have decreased behavioural plasticity in our modern breeds. Indeed, the study of reaction norms in wild fish and birds provides cautious evidence that behavioural plasticity (i.e., intra-individual variability) can have a heritable component [80,81]. And though a meta-analysis targeting the question of whether behaviour-selected working-breed dogs were more fixed in their behaviour than non-selected non-working breed dogs could not find any support for this hypothesis [31], it was reported that an unstable rearing environment does not seem to have an effect on the consistency of behaviour, further highlighting the stability of certain traits in breed dogs [73]. Exploring how this compares to free-ranging dogs is an exciting avenue for further research.

A last consideration regarding the test itself pertains to the exclusion of the play component in the Human approach test and what that teaches us about the need to consider one's study subject in all its facets. Play is frequently used in temperament tests to assess sociability [32,37,43], and the retainment of play behaviour into adulthood has also been suggested as a consequence of domestication in several species [reviewed in 45,82]. When applying this subtest with the street dogs, the low engagement rate and frequent fearful behaviour towards the toy might be explained by the negative or lack of experience such unowned free-ranging scavengers had with pulling on an object with a human [22]. Similarly, [18] suggested that free-ranging dogs' lower persistence in comparison to pet dogs was likely due to their need to preserve energy and a lack of experience with human-manipulated items. On the other hand, our

anecdotal observation with free-ranging populations indicates multiple incidents of conspecific and object play. Whether play behaviour does indeed persist into adulthood in free-ranging dogs in the conspecific context or solitary play, but not with a human partner, or whether our choice of method was just ill-considered, remains to explored and emphasizes how carefully the subject population, its ecology and life experience must be considered despite our extensive experience with the species itself.

Finally, two important limitations need to be addressed in terms of free-ranging dog testing as well as the representativeness of this study. First, the reliance on voluntary participation when testing adult free-ranging dogs creates a natural bias for more social animals, especially in longer and more interactive tests, and might be particularly challenging for feral populations (i.e., not reliant on human settlements). This bias is somewhat reduced in our Human approach test since the only selection criteria for starting a test was that the dog was seemingly alone. Hence, even the dogs that ran away fearfully upon approach were analysed as part of our dataset. The fact that we were able to retest more than half of these 'escaping' individuals and they manifested the same behaviour on second testing provides evidence that this measurement was indeed reliable. Nevertheless, these individuals were lost for all the following subtests and are therefore not part of the majority of the reliability analysis. At the same time, constraining a free-ranging animal would undoubtedly alter its natural behaviour in a test situation. Explorable options to test such dogs could be the use of shorter tests with little human presence wherever possible (e.g., reaction- or exploration-based tests) or testing populations that now live in a fenced shelter but genetically belong to the clade of free-ranging dogs and had spent most of their lives roaming freely. However, the influence of recent life experiences and social context in the shelter on the target behaviour have to be strongly considered [21]. Secondly, the fact that our study population has previously been recognized as exceptionally friendly [19] did not allow us to analyse the reliability of aggressive and many stress-related behaviours. For example, only 16.7% of our sample fled immediately and none showed aggression, while more than 50% of an Ethiopian village dog sample fled upon approach and 11% reacted aggressively (though one needs to consider that these dogs were not always tested alone or outside [30]). While expanding the presented test battery to other populations would initiate the characterisation of different free-ranging dog populations and what may affect differences in their behavioural profiles, we acknowledge that these limitations are likely difficult to be overcome. Extreme fear or the slightest sign of human-directed aggression will necessarily lead to aborting the test (particularly considering the risk of lethal disease, [e.g., 17]). And while using a fake dog rather than a real dog was indeed an attempt to test for the presence of aggression in a safe way, collecting additional observational data (regarding both human and conspecific directed behaviour) of the same individuals may be a complementary method allowing for multiple measures of specific behavioural traits.

In conclusion, the easy implementation, successful participation of most street dogs, and the high coding reliability presented here confirm that free-ranging dogs can be tested in longer test batteries with complex ethograms despite the challenging natural conditions. The wide variability in the test-retest reliability across subtests demonstrate that the management of disturbances, the choice of set up and test-retest intervals, and careful considerations of the subjects' life experience and ecology are central challenges when testing free-ranging dogs in their natural habitat. At the same time, the acceptable reliability in some of the subtest demonstrates that under well-chosen conditions, testing those populations is possible and opens a vast array of intriguing pathways to explore [22]. Bearing in mind the suggestions made in the discussion, going forward the test battery presented here can be a valuable method to test questions about cognitive traits and behavioural profiles in such an understudied population as free-ranging dogs.

## Supporting information

**S1 Video. Exemplary video showing the different subtests of the test battery.**
(MP4)

**S1 File.**
(ZIP)

**S2 File. Equipment information.**
(PDF)

**S1 Table. Detailed ethogram.**
(DOCX)

**S2 Table. Detailed inter- and intra-rater reliability results.**
(DOCX)

**S3 Table. Detailed test-retest reliability outcome.**
(DOCX)

## Acknowledgments

We thank Professor Ikhlass El Berbri for her support and collaboration in this project. Additionally, we thank the rest of our team in Morocco (in particular Andreas Berghänel, Haytem Bouchri, Manon Delaunay, and Magdalena Juskaite) without whom the project would not have been possible. We are also grateful to the population and authorities in the Sous-Massa region for supporting our field work. Finally, we want to extend our gratitude to the two reviewers for their valuable feedback and suggestions.

## Author Contributions

**Conceptualization:** Svenja Capitain, Giulia Cimarelli, Friederike Range, Sarah Marshall-Pescini.

**Data curation:** Svenja Capitain.

**Formal analysis:** Svenja Capitain.

**Funding acquisition:** Giulia Cimarelli, Friederike Range, Sarah Marshall-Pescini.

**Investigation:** Svenja Capitain, Giulia Cimarelli, Urša Blenkuš.

**Methodology:** Svenja Capitain, Urša Blenkuš, Sarah Marshall-Pescini.

**Project administration:** Svenja Capitain, Urša Blenkuš, Sarah Marshall-Pescini.

**Resources:** Giulia Cimarelli, Friederike Range, Sarah Marshall-Pescini.

**Supervision:** Giulia Cimarelli, Friederike Range, Sarah Marshall-Pescini.

**Validation:** Svenja Capitain.

**Visualization:** Svenja Capitain.

**Writing – original draft:** Svenja Capitain.

**Writing – review & editing:** Svenja Capitain, Giulia Cimarelli, Urša Blenkuš, Friederike Range, Sarah Marshall-Pescini.

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
