## [Decision Letter · Decision Letter 0]

12 Sep 2023

PONE-D-23-22870Tackling the lack of behavioural evidence for the domestication syndrome – Feasibility and reliability of a novel Domestication Syndrome test battery with free-ranging dogsPLOS ONE

Dear Dr. Capitain,

Thank you for submitting your manuscript to PLOS ONE. After careful consideration, we feel that it has merit but does not fully meet PLOS ONE’s publication criteria as it currently stands. Therefore, we invite you to submit a revised version of the manuscript that addresses the points raised during the review process.

We look forward to receiving your revised manuscript.

Kind regards,

Joshua Kamani, PhD

Academic Editor

PLOS ONE

3. We note that Figures 2 and 3 in your submission contain copyrighted images. All PLOS content is published under the Creative Commons Attribution License (CC BY 4.0), which means that the manuscript, images, and Supporting Information files will be freely available online, and any third party is permitted to access, download, copy, distribute, and use these materials in any way, even commercially, with proper attribution. For more information, see our copyright guidelines: http://journals.plos.org/plosone/s/licenses-and-copyright.

1. You may seek permission from the original copyright holder of Figures 2 and 3 to publish the content specifically under the CC BY 4.0 license.

Additional Editor Comments:

Dear Authors

Your manuscript has been assessed and serious flaws in the study design, DS concept and grammatical errors have been highlighted. Kindly and thoroughly address the issued raised by the reviews and resubmit for another assessment.

Thank you

Reviewers' comments:

Reviewer's Responses to Questions

**Comments to the Author**

1. Is the manuscript technically sound, and do the data support the conclusions?

Reviewer #1: Partly

Reviewer #2: No

2. Has the statistical analysis been performed appropriately and rigorously? 

Reviewer #1: Yes

Reviewer #2: I Don't Know

3. Have the authors made all data underlying the findings in their manuscript fully available?

Reviewer #1: Yes

Reviewer #2: Yes

4. Is the manuscript presented in an intelligible fashion and written in standard English?

Reviewer #1: Yes

Reviewer #2: Yes

5. Review Comments to the Author

Reviewer #1: GENERAL COMMENTS:

Let me start with saying that I am very happy to see a behaviour study on free-ranging dogs. However, I cannot recommend this study for publication in its current form for several reasons, the main one being that this is not a study of the domestication syndrome and I find its current presentation and narrative to be quite misleading in all honesty.

It is very unclear to me why the authors have chosen to present their manuscript so elaborate around the domestication syndrome when they a) do not confirm the presence of the domestication syndrome in the dog population (there is no test of whether the alleged domestication syndrome behaviours are in fact correlated – a central assumption of the hypothesis), and b) find that their so-called test battery fails to adequately quantify the behaviours they claim to be associated with the domestication syndrome. The eagerness to label their test battery the “Domestication Syndrome test battery” and claim that it is useful going forward in testing domestication syndrome behaviours, in spite of the very results they bring forward, therefore seem very misplaced and quite frankly puzzling. This is especially emphasized by the fact that half of this test battery includes behaviours that cannot in a meaningful way be associated with the domestication syndrome – a point the authors themselves hint at by mentioning that they will also test other domestication hypotheses with this test battery. These other behaviours, as compared to the claimed domestication syndrome behaviours, are even the only behaviours that with some confidence seem to be feasible and reliable to test in a free-ranging population. Why then label it the “Domestication Syndrome test battery”? It is simply misleading.

That being said, and while I understand that the above is a strong critique of the study, I think there is value in the study from the perspective that it is, in my opinion, actually a methods paper and it would benefit immensely from being rewritten as such. The work the authors have done in examining the feasibility and reliability in conducting in-situ behavioural testing of free-ranging dogs does have merit. So, instead of overselling this study by pinning it on a narrative that it simply cannot sustain, my strong recommendation for this manuscript is therefore that it undergoes major revision in order to be restructured to bring forward the valuable contributions it actually has in terms of the insights on general (not domestication syndrome) behavioural testing of free-ranging dogs. Highlighting the challenges of taking behavioural testing outside controlled laboratory settings and emphasize how this is a necessary next step in moving the field forward is valuable enough on its own.

DETAILED COMMENTS:

Line 29: The authors need to remove the physiological reference here. Even though hair samples were collected, they were not analyzed and we have no way of knowing if a) this method is reliable in obtaining physiological parameters, and b) if these correlate with the behavioural traits in the domestication syndrome. This reference is there for misleading in this context.

Line 53: Some clarity is needed here. Selection on tameness is not the mechanism per se. Tameness is likely the behavioural trait that was selected for during initial animal domestication cascading the phenotypic changes as seen in the domestication syndrome. However, the underlying mechanism causing the presumed correlated traits of the domestication syndrome remains unknown, but has been proposed to be neural crest deficit (Wilkins et al. 2016) or attenuated ontogenesis (Balyaev 1979).

Line 58: What are all the morphological domestication syndrome-associated traits? Details are generally very sparse and the background is too superficial.

Line 59: What physiological traits? Which traits are part of the DS and which do not fit with the hypothesis in dogs?

Line 60-61: Which behavioural traits? Which traits are part of the DS and which do not fit with the hypothesis in dogs?

Line 61- 66: This part has me a bit confused. It seems more like an afterthought and I am not quite sure how it fits in with the narrative. Are the authors saying that the domestication syndrome is not true in dogs or that it is expanded just in dogs with these extra features?

Line 64: I am quite surprised to see the uncritical and sole use of this outdated reference (Hare and Tomasello 2005), given the long-standing debate (of +20 years) on the topic and the not so few studies refuting the uniqueness in dogs being able to follow human communicative clues. Udell et al. (2008) demonstrated that wolves outperformed dogs in a pointing test when tested under identical conditions. Likewise, Wheat et al. (2022) reanalyzed previously published data claiming that dog puppies have early emerging skills in following human pointing whereas wolves do not, but upon adequately controlling for the puppies’ different environmental exposure dogs and wolves performed equally well. Furthermore, it has been demonstrated that close to 30 species (both wild and domesticated, e.g. elephants, bats, goats, pigs and not the least wolves) across eight taxonomic groups have the ability to follow pointing gestures (see Krause et al. 2018 for review).

Line 73-82: While I agree that it is problematic that some studies do not address the behavioural traits of the domestication syndrome in concert, I am puzzled by the argument that standardized test batteries designed to quantify behaviours, of which some are the basic behaviours of the domestication syndrome, should somehow be unfit for hypothesis testing. The argumentation brought forward by the authors here becomes quite ironic as they a) do not test all the behavioural traits of the domestication syndrome, and b) do not test the covariance between the behaviours they do test. There is thus no actual confirmation of the domestication syndrome in their free-ranging dog population, given one would accept the premise that the behaviours chosen are in fact domestication syndrome behaviours.

Line 83-94: I generally agree with this paragraph. However, it is too much of a blanket statement that free-ranging dogs in general is not admixed with breed dogs. For instance, Boyko et al. has a paper from 2009 showing that more than 60% of African village dogs could be classified as non-native, i.e. with a potential admixture with modern breeds. This does not mean that free-ranging dogs are not better models for these questions than modern breeds, but introgression with modern breeds must be assumed in free-breeding populations like this.

Line 100: I am starting to get a bit confused. Are the authors testing personality, the domestication syndrome or alterative domestication hypotheses?

Line 103-122: My confusion grows. Now it sounds more like a methods paper…

Line 124-126: If the battery test all these different hypotheses (which I do not believe it does, for instance there is no test of correlation between alleged domestication syndrome traits and the majority of domestication syndrome behaviours could not be adequately quantified) how come the authors label it the “Domestication Syndrome test battery”?

Line 145-146: This could create a sampling bias where only the tamest dogs participate, i.e. the most fearful dogs would not. This can ultimately affect results with exaggerated sociability scores and more fearful or less explorative dogs being underrepresented in the sample.

Line 160: Which domestication syndrome traits?

Line 193-203: This seems like a good test of human-directed sociability.

Line 225-253: Subtests 4-6, sure the authors wish to test other domestication hypotheses (though see my comments on pointing above), but since these tests take up half of the test battery and also seem to have the best feasibility and reliability, why then the eagerness to label the test battery the “Domestication Syndrome test battery”. It seems to me, that this test battery unlikely to actually quantify domestication syndrome behaviours, if at all.

Line 204-214: If the purpose is to quantify conspecific sociability and aggression, I am not sure I understand why the authors did not simply observe behaviours toward other dogs in the area? I am not convinced this test can be disentangled from neophobia and/or boldness.

Line 365-366: So, no aggression behaviour? Maybe one fear/stress/redirected behaviour (Yawning)?

Line 387: But I thought that tail between legs had poor interrater reliability and should be excluded (given that the authors adhere to the 0.07 treshold as stated in several places, e.g. line 345, 347 and 356).

Line 388: Could this low test-retest reliability be caused by habituation in spite modification of the fake dog. Maybe the testing set-up alone could be cause for habituation? Perhaps testing conspecific sociability and aggression is better in real life social contexts?

Line 401: Again, I am puzzled by the continued use of tail between legs even though it does not meet the authors stated 0.07 ICC threshold. I find it to be borderline misleading.

Line 404-405: This to me indicates both habituation, as mentioned above, but also that the fake dog was indeed treated as a novel object and not a suitable model of a conspecific.

Table 3: I find it very misleading, and confusing, to include variables that did not pass the ICC thresholds.

Line 467-475: I disagree with the interpretation of the results on sociability and the conclusions brought forward here. The sample presented here is very likely to be a result of the most social dogs in the population. This is because the dogs’ participation was entirely voluntarily and the dogs themselves could decide to leave the test at any given time. Since the study only includes test-retested dogs, this will inherently bias the sampling towards more social individuals and exclude fearful dogs.

Line 526-532: If the authors wish to speculate about phenotypic plasticity, I encourage them to do so in a more detailed way. What would the adaptive value of this be in free-ranging dogs?

Reviewer #2:

 There is too little known about the variation in behavior in stray/street dogs, and this study provides a survey of the types of behavioral tests that can and cannot be performed in this difficult testing environment. The study concept is thus very interesting and worthwhile. I enjoyed reading the description of traits tested and the results and discussion. As a preliminary study, I think it would be great to see the battery of tests published so that it could be retested with a larger sample.

However, I think the title, hypotheses, and design of the paper show a fundamental misunderstanding of some of the basic concepts in domestication studies. Domestication and the Domestication Syndrome (DS) hypothesis are not equivalent concepts, and the two are confounded repeatedly throughout the paper. The DS is a well-known and highly debated hypothesis that has been tested in many ways, taken apart, and found to be highly problematic. Most domestication workers agree that is has a very specific applicability when being tested. I provided more details on relevant sources in my "Itemized Review".There is no discussion of these issues in the present manuscript. The DS aims to explain the generalized traits co-ocurring in domestic populations when compared to wild counterparts. Dogs have repeatedly been criticized as a bad model for testing "early domestication" or the DS, due to their artificial breeding. In my opinion your study does not test the DS. It tests the presence of certain human-dog interactions that reflect tameness or lack of aggression in stray dogs. It is successful in that, and would be a great survey to publish as a starting point for behavioral studies in street dogs. However, as stated in my Itemized Review, the DS is not tested here. Tameness is but one trait, and it is present in all domesticated populations that have been studied thus far. I believe the authors could redesign and re-work this study with a better understanding of what they are testing. My best wishes to you and this work. I think it would be great to see this published once it is improved.

6. PLOS authors have the option to publish the peer review history of their article (what does this mean?). If published, this will include your full peer review and any attached files.

Reviewer #1: No

Reviewer #2: No

---

## [Author Response · Author response to Decision Letter 0]

27 Oct 2023

Additional Editor Comments:

Dear Authors

Your manuscript has been assessed and serious flaws in the study design, DS concept and grammatical errors have been highlighted. Kindly and thoroughly address the issued raised by the reviews and resubmit for another assessment.

Thank you

We are very grateful for the reviewers’ detailed clarifications and suggestions, as well as their encouraging words and their concern with the description of the domestication-related focus. We agree with the suggestion that changing the focus of this paper to free-ranging dog testing overall provides more benefit for the scientific community and the reader. Following both reviewers’ suggestion, we have rewritten the paper by dropping the domestication syndrome constraints and refocussing on the merits this paper has. 

The requirements have been rechecked and implemented. Particularly file naming for figures and supplementary material has been corrected. 

2. Please include a complete copy of PLOS’ questionnaire on inclusivity in global research in your revised manuscript. 

The requested file has been uploaded.

3. We note that Figures 2 and 3 in your submission contain copyrighted images. We require you to either (1) present written permission from the copyright holder to publish these figures specifically under the CC BY 4.0 license, or (2) remove the figures from your submission. 

Figure 2 has been remade to only contain MS PowerPoint pictograms which are free to use. Figure 3 has been removed and links to the specific manufacturers have been included in the supplementary information S2 in case visual information is desired. 

Reviewer 1

Reviewer #1: GENERAL COMMENTS:

Let me start with saying that I am very happy to see a behaviour study on free-ranging dogs. However, I cannot recommend this study for publication in its current form for several reasons, the main one being that this is not a study of the domestication syndrome and I find its current presentation and narrative to be quite misleading in all honesty.

It is very unclear to me why the authors have chosen to present their manuscript so elaborate around the domestication syndrome when they a) do not confirm the presence of the domestication syndrome in the dog population (there is no test of whether the alleged domestication syndrome behaviours are in fact correlated – a central assumption of the hypothesis), and b) find that their so-called test battery fails to adequately quantify the behaviours they claim to be associated with the domestication syndrome. The eagerness to label their test battery the “Domestication Syndrome test battery” and claim that it is useful going forward in testing domestication syndrome behaviours, in spite of the very results they bring forward, therefore seem very misplaced and quite frankly puzzling. This is especially emphasized by the fact that half of this test battery includes behaviours that cannot in a meaningful way be associated with the domestication syndrome – a point the authors themselves hint at by mentioning that they will also test other domestication hypotheses with this test battery. These other behaviours, as compared to the claimed domestication syndrome behaviours, are even the only behaviours that with some confidence seem to be feasible and reliable to test in a free-ranging population. Why then label it the “Domestication Syndrome test battery”? It is simply misleading.

That being said, and while I understand that the above is a strong critique of the study, I think there is value in the study from the perspective that it is, in my opinion, actually a methods paper and it would benefit immensely from being rewritten as such. The work the authors have done in examining the feasibility and reliability in conducting in-situ behavioural testing of free-ranging dogs does have merit. So, instead of overselling this study by pinning it on a narrative that it simply cannot sustain, my strong recommendation for this manuscript is therefore that it undergoes major revision in order to be restructured to bring forward the valuable contributions it actually has in terms of the insights on general (not domestication syndrome) behavioural testing of free-ranging dogs. Highlighting the challenges of taking behavioural testing outside controlled laboratory settings and emphasize how this is a necessary next step in moving the field forward is valuable enough on its own.

GENERAL ANSWER to Reviewer #1: 

Thank you for your valuable perspective and detailed concerns regarding the approach we had chosen to present this study in. The paper was indeed supposed to be a methods paper and we regret that it did not come across as such. Taking your suggestions to heart, it has been entirely rewritten, focusing on the necessity of testing free-ranging populations, how the presented test battery fits into that, and what conclusions can be drawn from our results about the possibilities and limitations of street dog testing. This pertains particularly to the introduction and partially to the discussion, where we have additionally addressed issues you raised with the methodology. The methods and results section have stayed largely the same besides clarifying specific issues you have pointed out, and renaming the Sociability test as Human approach test to clarify that all behaviours were tested. We hope that by refocussing this paper on the merits it has in terms of moving the field forward, this version is more to your satisfaction. 

DETAILED COMMENTS:

Line 29: The authors need to remove the physiological reference here. Even though hair samples were collected, they were not analyzed and we have no way of knowing if a) this method is reliable in obtaining physiological parameters, and b) if these correlate with the behavioural traits in the domestication syndrome. This reference is there for misleading in this context.

Our answer: With the rewriting of the abstract, this line was taken out. 

Line 53: Some clarity is needed here. Selection on tameness is not the mechanism per se. Tameness is likely the behavioural trait that was selected for during initial animal domestication cascading the phenotypic changes as seen in the domestication syndrome. However, the underlying mechanism causing the presumed correlated traits of the domestication syndrome remains unknown, but has been proposed to be neural crest deficit (Wilkins et al. 2016) or attenuated ontogenesis (Balyaev 1979).

Our answer: As expressed above, we want to thank you for your clarifications and suggestions particularly in terms of refocusing the paper. Since the introduction has been entirely rewritten in this effort, many of the issues you have pointed out in this comment and below have been erased from the text. Whenever this is the case, we will mark it with “erased due to focus shift”. 

Line 58: What are all the morphological domestication syndrome-associated traits? Details are generally very sparse and the background is too superficial.

Our answer: “erased due to focus shift”

Line 59: What physiological traits? Which traits are part of the DS and which do not fit with the hypothesis in dogs?

Our answer: “erased due to focus shift”

Line 60-61: Which behavioural traits? Which traits are part of the DS and which do not fit with the hypothesis in dogs?

Our answer: “erased due to focus shift”

Line 61- 66: This part has me a bit confused. It seems more like an afterthought and I am not quite sure how it fits in with the narrative. Are the authors saying that the domestication syndrome is not true in dogs or that it is expanded just in dogs with these extra features?

Our answer: “erased due to focus shift”

Line 64: I am quite surprised to see the uncritical and sole use of this outdated reference (Hare and Tomasello 2005), given the long-standing debate (of +20 years) on the topic and the not so few studies refuting the uniqueness in dogs being able to follow human communicative clues. Udell et al. (2008) demonstrated that wolves outperformed dogs in a pointing test when tested under identical conditions. Likewise, Wheat et al. (2022) reanalyzed previously published data claiming that dog puppies have early emerging skills in following human pointing whereas wolves do not, but upon adequately controlling for the puppies’ different environmental exposure dogs and wolves performed equally well. Furthermore, it has been demonstrated that close to 30 species (both wild and domesticated, e.g. elephants, bats, goats, pigs and not the least wolves) across eight taxonomic groups have the ability to follow pointing gestures (see Krause et al. 2018 for review).

Our answer: “erased due to focus shift”. The reference still appears in the manuscript but only as a reference to Hare’s emotional reactivity hypothesis. In addition, we want to thank you for pointing us to this useful reference. We included Krause’s observations in line 214.

Line 73-82: While I agree that it is problematic that some studies do not address the behavioural traits of the domestication syndrome in concert, I am puzzled by the argument that standardized test batteries designed to quantify behaviours, of which some are the basic behaviours of the domestication syndrome, should somehow be unfit for hypothesis testing. The argumentation brought forward by the authors here becomes quite ironic as they a) do not test all the behavioural traits of the domestication syndrome, and b) do not test the covariance between the behaviours they do test. There is thus no actual confirmation of the domestication syndrome in their free-ranging dog population, given one would accept the premise that the behaviours chosen are in fact domestication syndrome behaviours.

Our answer: “erased due to focus shift”. In addition, a) is a very fair argument and a major reason why we agreed to drop the Domestication Syndrome narrative. For point b), analyses might follow in the future if the test can be established reliably on the streets, which was the main focus of this paper. 

Line 83-94: I generally agree with this paragraph. However, it is too much of a blanket statement that free-ranging dogs in general is not admixed with breed dogs. For instance, Boyko et al. has a paper from 2009 showing that more than 60% of African village dogs could be classified as non-native, i.e. with a potential admixture with modern breeds. This does not mean that free-ranging dogs are not better models for these questions than modern breeds, but introgression with modern breeds must be assumed in free-breeding populations like this.

Our answer: Thank you for this useful reference, it was a pleasure to read. Boyko et al. (2009) do indeed state that village dogs in some areas of Africa (esp. Namibia) show a high admixture rate while this admixture was low or non-existent in the other 84% of their sample. We have softened our statement (see line 84ff) to indicate that the “the majority of free-ranging dog populations are not a mixture of breeds but rather genetically distinct populations that predate the relatively more recent artificial selection and breed formation”, and refer to Boyko et al. (2009). 

Line 100: I am starting to get a bit confused. Are the authors testing personality, the domestication syndrome or alterative domestication hypotheses?

Our answer: “erased due to focus shift”

Line 103-122: My confusion grows. Now it sounds more like a methods paper…

Our answer: That is indeed what it was supposed to be. We hope this version is more to your satisfaction. 

Line 124-126: If the battery test all these different hypotheses (which I do not believe it does, for instance there is no test of correlation between alleged domestication syndrome traits and the majority of domestication syndrome behaviours could not be adequately quantified) how come the authors label it the “Domestication Syndrome test battery”?

Our answer: “erased due to focus shift”

Line 145-146: This could create a sampling bias where only the tamest dogs participate, i.e. the most fearful dogs would not. This can ultimately affect results with exaggerated sociability scores and more fearful or less explorative dogs being underrepresented in the sample.

Our answer: That is indeed an important limitation of street tests since forcing the animals to participate would likely alter their behaviour even more, and we thus have to contend with the voluntary biased sample. While the human approach test somewhat circumvents this problem by starting the data collection before the dog is aware of being approached, we have to concede that this is a problem for all the other following subtests. We have endeavoured to address this point in the discussion (line 583ff) and mention a few possible alternatives there as well. 

Line 160: Which domestication syndrome traits?

Our answer: “erased due to focus shift”

Line 193-203: This seems like a good test of human-directed sociability.

Our answer: We are glad you agree. 

Line 225-253: Subtests 4-6, sure the authors wish to test other domestication hypotheses (though see my comments on pointing above), but since these tests take up half of the test battery and also seem to have the best feasibility and reliability, why then the eagerness to label the test battery the “Domestication Syndrome test battery”. It seems to me, that this test battery unlikely to actually quantify domestication syndrome behaviours, if at all.

Our answer: “erased due to focus shift”

Line 204-214: If the purpose is to quantify conspecific sociability and aggression, I am not sure I understand why the authors did not simply observe behaviours toward other dogs in the area? I am not convinced this test can be disentangled from neophobia and/or boldness.

Our answer: Thank you for stating this concern. We do agree that elaborate long-term observations are the best way to assess a dog’s social behaviour, whether that is pet, working, or free-ranging dog. However, this test was conceptualized to circumvent the necessity of elaborate observations to create the possibility to test more dogs without having to learn everything about their population first. In addition, we aimed at exploring the possibility of using established pet and working dog test batteries on the streets. The fake dog test has repeatedly been validated to reliably assess dogs’ initial behaviour towards conspecifics (Barnard et al., 2019; Reid et al., 2022) and we were therefore positive that the same could be achieved on the street. With the outcome that the street dogs did indeed show appropriate conspecific behaviour (e.g., growling, barking, stand tall) upon first interaction with the fake dog that was not seen in the novel object test, as well as frequent genital sniffing, we argue that the test did achieve that goal and can be used to assess the target measures when only the initial reaction is taken into account. Nevertheless, we cannot exclude that neophobia and boldness come into play as soon as the animals realize that the opponent is not, in fact, a dog. We discuss this issue in line 528f as well as the value of observational data in line 614f. 

Line 365-366: So, no aggression behaviour? Maybe one fear/stress/redirected behaviour (Yawning)?

Our answer: Indeed, the aggressive behaviours and many stress behaviours were rarely shown and could thus not be retained in the test-retest analysis. We discuss the problems with that as well as possible alternatives in line 602ff. 

Line 387: But I thought that tail between legs had poor interrater reliability and should be excluded (given that the authors adhere to the 0.07 threshold as stated in several places, e.g. line 345, 347 and 356).

Our answer: We apologize for this confusion. The tail between the leg and mouthing behaviour were close to the threshold and showed marked significance (p=0.02). After long discussion and in light of the somewhat arbitrary 0.7 threshold that is heavily influenced by sample size, we chose to retain the two variables for this preliminary investigation and treat them with caution. However, we do agree that we should have stated that more clearly and understand your annoyance with the lack thereof. An explanation has been added in the methods (line 299f) and the result section (line 356ff). 

Line 388: Could this low test-retest reliability be caused by habituation in spite modification of the fake dog. Maybe the testing set-up alone could be cause for habituation? Perhaps testing conspecific sociability and aggression is better in real life social contexts?

Our answer: Absolutely. We share this interpretation wholeheartedly, as we have explained in line 519f. Pitfalls of short retest intervals and too similar set ups are described in the paragraph thereafter (lines 522ff), and we have added the value of long-term observations in line 614. 

Line 401: Again, I am puzzled by the continued use of tail between legs even though it does not meet the authors stated 0.07 ICC threshold. I find it to be borderline misleading.

Our answer: Understandably so, we apologize for this lack of explanation. Please refer to our explanation for line 387 above for our modifications. 

Line 404-405: This to me indicates both habituation, as mentioned above, but also that the fake dog was indeed treated as a novel object and not a suitable model of a conspecific.

Our answer: We agree with this interpretation and have discussed the limitations as mentioned above. 

Table 3: I find it very misleading, and confusing, to include variables that did not pass the ICC thresholds.

Our answer: Again, we apologize and agree that we should have indicated this in a better way. As mentioned above, respective measures have been taken to do so. 

Line 467-475: I disagree with the interpretation of the results on sociability and the conclusions brought forward here. The sample presented here is very likely to be a result of the most social dogs in the population. This is because the dogs’ participation was entirely voluntarily and the dogs themselves could decide to leave the test at any given time. Since the study only includes test-retested dogs, this will inherently bias the sampling towards more social individuals and exclude fearful dogs.

Our answer: Thank you for raising this concern. You are entirely correct that the voluntary participation creates a biased sample for the overall test-battery, and we have added a paragraph in lines 583ff to address this issue. At the same time, we would like to draw attention to three points that might alleviate your concern a little in this specific instance, i.e., the sociability (now Human approach) subtest. 1) As it was the very first subtest, the animals were not first required to come to us but rather we started the approach for this subtest before the animals were aware of it. More precisely, we appeared as normal locals to the animal right until I started the (now Human approach) test by calling their attention to me with a short vocalization. Since many animals were foraging or sleeping up to that point, they did not have a chance to flee beforehand. That means that by the time the very fearful animals started to flee, the test had already started, and their behaviour was still recorded and coded as part of the (now Human approach) test. 2) to address your concern about a bias in the retesting, we did find two-thirds of the dogs again that had already fled during the (now Human approach) subtest in the first test, creating a decent sample of fearful animals that is retained in the test-retest analysis for the sociability test. 3) And finally, the fact that only the more social animals stuck around during the tests does not negate the fact that sociability might be heritable. We do not mean to claim here that street dogs are more social than wolves (since we do not have that data and the skew would indeed cause a big issue with that) but rather that the sociability measure seems to be fairly consistent over time in an individual and could therefore serve as a trait based on which selection is possible. Rereading the sentence now I can see where it might have been interpreted differently and we have specified that we refer to consistency (line 481). To additionally counteract the perceived bias during this test, we renamed it to a more neutral “Human approach test” rather than the more directed “Sociability test”. 

Line 526-532: If the authors wish to speculate about phenotypic plasticity, I encourage them to do so in a more detailed way. What would the adaptive value of this be in free-ranging dogs?

Our answer: Thank you for this encouragement. We have added a few lines how behavioural plasticity might aid free-ranging dogs in adapting to their unstable and variable environment, which is not a necessity for the average breed dog anymore (lines 550f). 

GENERAL COMMENT Reviewer 2

There is too little known about the variation in behavior in stray/street dogs, and this study provides a survey of the types of behavioral tests that can and cannot be performed in this difficult testing environment. The study concept is thus very interesting and worthwhile. I enjoyed reading the description of traits tested and the results and discussion. As a preliminary study, I think it would be great to see the battery of tests published so that it could be retested with a larger sample.

However, I think the title, hypotheses, and design of the paper show a fundamental misunderstanding of some of the basic concepts in domestication studies. Domestication and the Domestication Syndrome (DS) hypothesis are not equivalent concepts, and the two are confounded repeatedly throughout the paper. The DS is a well-known and highly debated hypothesis that has been tested in many ways, taken apart, and found to be highly problematic. Most domestication workers agree that is has a very specific applicability when being tested. I provided more details on relevant sources in my "Itemized Review".There is no discussion of these issues in the present manuscript. The DS aims to explain the generalized traits co-ocurring in domestic populations when compared to wild counterparts. Dogs have repeatedly been criticized as a bad model for testing "early domestication" or the DS, due to their artificial breeding. In my opinion your study does not test the DS. It tests the presence of certain human-dog interactions that reflect tameness or lack of aggression in stray dogs. It is successful in that, and would be a great survey to publish as a starting point for behavioral studies in street dogs. However, as stated in my Itemized Review, the DS is not tested here. Tameness is but one trait, and it is present in all domesticated populations that have been studied thus far. I believe the authors could redesign and re-work this study with a better understanding of what they are testing. My best wishes to you and this work. I think it would be great to see this published once it is improved.

GENERAL ANSWER TO REVIEWER #2:

We would like to thank you for your helpful clarifications regarding the domestication syndrome and detailed elaborations how this paper could be improved. We agree that the focus of this paper was not ideally represented. Taking your encouragement and the suggestion of reviewer #1 into account, we chose to entirely refocus the paper towards the possibilities and limitations of free-ranging dog testing and centre our study within that. Since we do not have a large enough sample to reliably represent the distribution of specific traits across the population (as you mention, a larger sample would be needed for that), we solely focus on the presentation and reliability validation of the presented test battery to set a starting point for more elaborate testing. In addition, we discuss important considerations our results illustrate and alternatives that might be viable. These changes can be found especially in the entirely rewritten introduction and partially rewritten discussion, while the methods and result section stayed largely the same. Finally, we defined and used the terms free-ranging and street dogs more carefully and took your suggested references on board wherever possible, thank you for those pointers. We hope that by refocussing this paper on the merits it has in terms of moving the field forward, this version is more to your satisfaction.

Abstract:

A brief explanation about what is meant by “human-directed interactions” vs. “disturbance-prone behaviors” vs. “less disturbance-prone” would help the reader understand your study better from the very beginning. 

Our answer: Thank you for pointing this out. We have entirely rewritten the abstract and therein explained the disturbance issue with more detail.

The term “free-ranging” is a bit vague and can be misinterpreted unless you define it specifically. For example, feral dogs like the Dingo have been described as free-ranging. Can your population perhaps be described further as “street dogs”? This is an important item because the type of population you are using relates directly to the testability of your hypothesis.

Our answer: This explanation was very helpful, thank you for that. We have added a definition for free-ranging dogs in the manuscript (line 57f), and while we continue to refer to free-ranging dogs wherever appropriate, we specifically state our use of street dogs for the test battery and adapted our interpretations and discussions to refer to the population our results can or cannot be generalized to. 

Lines 44-45: the DS is not an attempt to explain domestication, but the traits associated with domestication. This may seem like a trivial difference but is really important to making

your study more scholarly. 

Our answer: That is indeed an important differentiation. While we have erased the words pertaining to this from the ms, we will endeavour to be more accurate in our future work.

Introduction:

Lines 43-44: The methods/pathways used to domesticate animals are pretty well known (see Zeder 2012 for a good overview). It is the mechanisms by which multiple morphological/phenotypic traits correlated with domestication are co-occurring repeatedly that is misunderstood. Again, I think this is an important distinction to make throughout your paper. Domestication is something that researchers struggle to define in a simple manner—but there is much literature on this—so defining the “type” of domestication you are testing is critical. I strongly suggest that you read Zeder 2012 for and overview of the different types of processes involved in the different types of “domestication”. Another view of what constitutes a “domestic” population is that of Lord et al. (several) – so there is much debate. This will inform whether your sampling is appropriate or not. 

Our answer: Thank you for the reference suggestions. While we have erased the words pertaining to this from the ms, this is certainly a note we will keep in mind. 

Line 56: Brain size reduction is not inconsistent among domesticated groups—quite the opposite. You can check Balcarcel et al. 2021 for a review of the literature on this subject and references therein, or Kruska’s work from the 1970s-2000’s. Your study cites Johnsson et al. 2021, which is a great paper, but I do believe there is an error in it regarding the statement of “increase” in brain size in pigs, llamas and alpacas. Johnsson et al cite Kruska (1974, 1982, 1988) and if you look at those original works, they report decreases in the telencephalon (not increases). 

Our answer: That is an interesting thought and fact I was not aware of, thank you. Again something to keep in mind, though the line itself does not apply to the paper anymore. 

Lines 59-60: There is a tremendous amount of literature debating the claims of the DS, its unsuitability in describing the variation presented by different domestic populations. I would suggest discussing this a bit, to show your understanding of the subject, before getting into testing of the DS.

a. For example, some domestication workers agree that dogs are a poor model for testing the DS hypothesis, to due to their directed artificial breeding during the last 200 yrs. (see Larson et al. 2014, McHugh & Larson 2017, Sanchez et al. 2016&2017 on lack of a universal pattern among domestics and the NCC).

b. Dogs also don’t necessarily show snout shortening, as claimed by the DS. Please see Drake et al. 2010 on the variation in cranio-facial changes in dog breeds which highlights the shortening of the snout as an artificially-selected trait in many breeds. 

Our answer: Thank you for this suggestion and we agree wholeheartedly, hence our suggestion that free-ranging dogs might be a better population. While most of this is not part of the ms anymore, we have incorporated some of it in line 80ff. Your references were very helpful in that. 

IN GENERAL: You may want to rephrase or discuss the intricacies of the dog as a model for testing evolutionary processes. There is great potential in this model group, but not necessarily for testing the DS, in my opinion, since the DS is supposed to capture basic domestication traits common to all domestic populations…and dogs are highly artificially-selected. I think your study actually tests a *much broader* concept in domestication—the selection for tameness and reduced aggression—which is tremendously interesting.

Our answer: That is a valuable suggestion and with the rewriting of this ms we have attempted to shift the focus away from the ms and incorporated more broader domestication ideas particularly in the discussion (e.g., lines 481ff)

Lines 69-71: Here again, I think the authors may be confounding the domestication with "domestication syndrome". 

Our answer: This line has been erased from the ms

“Free-ranging”. I am concerned about the use of this term, and what it means for your sample. Hughes and MacDonald 2013 do not exclude feral populations under their definition of free-roaming dogs. Pilot et al. 2015 cite Hughes and MacDonald 2013, adding "not feral" to the definition, but I am not sure if there is evidence for that? 

Our answer: Please see point 2 under abstract for our answer and implemented changes in regards to that.

---

## [Decision Letter · Decision Letter 1]

24 Nov 2023

PONE-D-23-22870R1Street-wise dog testing: feasibility and reliability of a behavioural test battery for free-ranging dogs in their natural habitat

Dear Dr. Capitain,

Thank you for submitting your manuscript to PLOS ONE. After careful consideration, we feel that it has merit but does not fully meet PLOS ONE’s publication criteria as it currently stands. There are some issues you need to address to improve the manuscript. Therefore, we invite you to submit a revised version of the manuscript that addresses the points raised during the review process.Thank you

Please submit your revised manuscript by Jan 08 2024 11:59PM. If you will need more time than this to complete your revisions, please reply to this message or contact the journal office at plosone@plos.org. Please include the following items when submitting your revised manuscript:A rebuttal letter that responds to each point raised by the academic editor and reviewer(s). You should upload this letter as a separate file labeled 'Response to Reviewers'.A marked-up copy of your manuscript that highlights changes made to the original version. You should upload this as a separate file labeled 'Revised Manuscript with Track Changes'.An unmarked version of your revised paper without tracked changes. You should upload this as a separate file labeled 'Manuscript'.If applicable, we recommend that you deposit your laboratory protocols in protocols.io to enhance the reproducibility of your results. Protocols.io assigns your protocol its own identifier (DOI) so that it can be cited independently in the future. For instructions see: https://journals.plos.org/plosone/s/submission-guidelines#loc-laboratory-protocols. Additionally, PLOS ONE offers an option for publishing peer-reviewed Lab Protocol articles, which describe protocols hosted on protocols.io. Read more information on sharing protocols at https://plos.org/protocols?utm_medium=editorial-email&utm_source=authorletters&utm_campaign=protocols.

We look forward to receiving your revised manuscript.

Kind regards,

Joshua Kamani, PhD

Academic Editor

PLOS ONE

Journal Requirements:

Reviewers' comments:

Reviewer's Responses to Questions

**Comments to the Author**

1. If the authors have adequately addressed your comments raised in a previous round of review and you feel that this manuscript is now acceptable for publication, you may indicate that here to bypass the “Comments to the Author” section, enter your conflict of interest statement in the “Confidential to Editor” section, and submit your "Accept" recommendation.

Reviewer #1: All comments have been addressed

Reviewer #2: (No Response)

2. Is the manuscript technically sound, and do the data support the conclusions?

Reviewer #1: Yes

Reviewer #2: Yes

3. Has the statistical analysis been performed appropriately and rigorously? 

Reviewer #1: Yes

Reviewer #2: I Don't Know

4. Have the authors made all data underlying the findings in their manuscript fully available?

Reviewer #1: Yes

Reviewer #2: Yes

5. Is the manuscript presented in an intelligible fashion and written in standard English?

Reviewer #1: Yes

Reviewer #2: Yes

6. Review Comments to the Author

Reviewer #1: Thank you for giving me the opportunity to review the revised version of this manuscript. I really appreciate the great effort the authors have put into changing the narrative to a methods paper. While I think the manuscript is much improved, I, however, still have some comments that I feel could still substantially improve the submission.

My main issue with the manuscript in its current form is that the great clarity and conciseness of the abstract is somewhat lost in the main text, where especially the introduction, and to some extend the discussion, get overly wordy and complicated. This is a shame, as the message in the abstract is so to the point and all the extra fill added in the manuscript text takes away from this, leaving the reader with a quite confused impression in which the developed methods end up coming across much vaguer than they actually are.

I therefore mainly have general comments regarding this issue and suggestions on how to deal with them:

INTRODUCTION

The narrative about changes during domestication still seems to be forced into the manuscript and it is distracting from what it is really about: a methods paper testing whether it is feasible to use a behavioral test battery on free-ranging dogs (like it is used on pet dogs). For instance, in the very first line in the introduction (line 45-47) it is mentioned which behavioral traits are expected to change during dog domestication but this is irrelevant to the narrative of the sentence and the rest of the paragraph. Why not just focus on the history of dog domestication and the interest of dogs in science?

This line of argumentation leads me to the paragraph spanning line 76-92. I have already stated in my previous review that I largely agree that pet/breed dogs are not the best lineages of dogs for questions relating to domestication. This point has been emphasized by several large-scale genetic and behavioral studies. However, I do have to object to the statement that free-ranging dogs per se are better candidates for studying dog domestication. I suspect that the authors might be conflating the generic term free-ranging dogs with village dogs, which are independent canid lineages that predates breed dog formation. While true that the majority of the worlds’ dog population is indeed free-ranging, all free-ranging dogs are not village dogs. A significant proportion of free-ranging dogs are abandoned pet dogs and should not automatically be labeled as an independent genetic lineage without some sort of genetic evidence hereof. I also feel that the Boyko references is particularly misguided in its use in line 86 as they show that while authentic village dog lineages do exist in Africa, many dogs are heavily admixed with breed dog lineages.

This is not to say that I disagree with the argumentation that 1) free-ranging dog populations are worth studying (they are and we should), and 2) that some of them are better representatives than breed dogs. I just think that especially point 2) should be approached with caution and maybe in this context, where the authors do not in fact have any evidence for the genetic makeup of their free-ranging dog population, should err on the side of the narrative of point 1) and simply argue that because the vast majority of the worlds dog population is free-ranging AND thus free of strong artificial selection, we should simply make the effort to develop validated methods to quantify the behaviors of these dogs. That alone is a strong and clear argument – just like in the abstract. The other narrative (i.e. point 2)) is not important to the methods presented and just adds clutter and confusion. Line 93-120 is what it is all about – rigor and repeatability, love to see it!

Specific comments:

Line 112: Interchanging use of free-ranging dogs and street dogs throughout the manuscript, which is a bit confusing.

Line 114: Which domestication hypotheses. Why is this relevant to this methods paper?

Line 186: Seems like the authors are testing multiple behaviors at the same time here. Playfulness could be independent of sociability, no? Will sociability and play be separated in the analyses? Ah, I can see that playfulness was dropped from the analyses…

Line 187: My objections to using the fake dog as a test of conspecific aggression from the first review remain unchanged given the results presented (even though the issues with the test possibly quantifying neophobia and boldness instead of conspecific aggression is now lightly touched upon in line 530-532).

Line 211: Please also see Nawroth, C., Wiesmann, K., Schlup, P. et al. Domestication and breeding objective did not shape the interpretation of physical and social cues in goats (Capra hircus). Sci Rep 13, 19098 (2023) https://doi.org/10.1038/s41598-023-46373-9 for the latest on the pointing debate.

Line 479-483: Not necessarily. These dogs could also have learned that humans mean food, which seems highly likely given that they are all scavengers. So, the behaviors the authors here associate with consistency and heritability could simply be the result of a learned behavioral process. How can the authors disentangle these concepts when they do not know the background (genetic, social etc.) of the dogs they are testing? This part is very speculative at best.

Line 502-511: This is a thoughtful paragraph. I wonder if point 3) does not leave room for habituation to the test situation, which might affect behavioral measurements?

Line 565-566: The authors have not introduced the domestication syndrome until this point. The reader does not know that this syndrome, or neoteny, is and it is not relevant to the narrative. Strongly suggest deleting this part.

Line 571-575: This part is only relevant if the authors know for sure that their dog population predates breed formation and they currently do not have the genetic evidence to back this statement up. In this light I find this line of argumentation irrelevant and speculative in regard to the results presented.

Reviewer #2: I thank the Authors for addressing previous comments.

The scope is much more constrained and I find it has improved the work significantly, resulting

in a methods paper that domestication workers will find useful in the design of larger studies, and very

informative on the challenges of behavioral studies outside of laboratory environments.

My full comments are included in a separate word document. Please see the manuscript pdf, however, which is where I tracked the grammatical edits, of which there were numerous. I took the liberty of suggesting better word structure, but these are simply my suggestions, of course.

7. PLOS authors have the option to publish the peer review history of their article (what does this mean?). If published, this will include your full peer review and any attached files.

Reviewer #1: No

Reviewer #2: **Yes: **Ana Balcarcel

---

## [Author Response · Author response to Decision Letter 1]

29 Nov 2023

Reviewer #1:

Reviewer #1: Thank you for giving me the opportunity to review the revised version of this manuscript. I really appreciate the great effort the authors have put into changing the narrative to a methods paper. While I think the manuscript is much improved, I, however, still have some comments that I feel could still substantially improve the submission.

My main issue with the manuscript in its current form is that the great clarity and conciseness of the abstract is somewhat lost in the main text, where especially the introduction, and to some extend the discussion, get overly wordy and complicated. This is a shame, as the message in the abstract is so to the point and all the extra fill added in the manuscript text takes away from this, leaving the reader with a quite confused impression in which the developed methods end up coming across much vaguer than they actually are. I therefore mainly have general comments regarding this issue and suggestions on how to deal with them:

General answer to Reviewer #1: 

We would like to take this opportunity to thank you for your valuable thoughts and explaining your point of view in detail. We agree with the majority of your points and have taken them on board (detailed below). Therein, particularly the first half of the introduction has been slimmed down and the domestication subject has been reduced, though we do argue for its merit and hence retained a more nuanced picture of it. We hope that this version makes for an easier and more satisfying read and does the topic justice.

INTRODUCTION

The narrative about changes during domestication still seems to be forced into the manuscript and it is distracting from what it is really about: a methods paper testing whether it is feasible to use a behavioral test battery on free-ranging dogs (like it is used on pet dogs). 

Answer: Thank you for seeing the merit in our paper and your detailed suggestions. We do agree that the domestication subject can be dropped in some areas (detailed below). However, we believe that disregarding the value free-ranging dogs could have for domestication research would not do the topic justice. And since reviewer 2 was delighted about the inclusion of this topic ("I am certain that the study adds great value to domestication studies focused on behavioral differences across populations"), we have chosen to retain it as a point of focus. Nevertheless, your arguments were convincing in that our iteration of the topic was not differentiated enough and took up too much space. Hence, we have cut it down in line with your suggestions and only retain it as one of the arguments for studying free-ranging populations, including the needed differentiation of breed-admixture in free-roaming populations. 

For instance, in the very first line in the introduction (line 45-47) it is mentioned which behavioral traits are expected to change during dog domestication but this is irrelevant to the narrative of the sentence and the rest of the paragraph. Why not just focus on the history of dog domestication and the interest of dogs in science?

Answer: L45: the part about specific hypotheses has been removed to improve clarity. To underline for the lay person why behaviour might have been of interest to humans since early on, we did change the second part of the sentence to simply point to human-directed selection, though excluding the points to the specific hypotheses. 

This line of argumentation leads me to the paragraph spanning line 76-92. I have already stated in my previous review that I largely agree that pet/breed dogs are not the best lineages of dogs for questions relating to domestication. This point has been emphasized by several large-scale genetic and behavioral studies. However, I do have to object to the statement that free-ranging dogs per se are better candidates for studying dog domestication. I suspect that the authors might be conflating the generic term free-ranging dogs with village dogs, which are independent canid lineages that predates breed dog formation. While true that the majority of the worlds’ dog population is indeed free-ranging, all free-ranging dogs are not village dogs. A significant proportion of free-ranging dogs are abandoned pet dogs and should not automatically be labeled as an independent genetic lineage without some sort of genetic evidence hereof. I also feel that the Boyko references is particularly misguided in its use in line 86 as they show that while authentic village dog lineages do exist in Africa, many dogs are heavily admixed with breed dog lineages.

Answer: L76-92: Thank you for pointing out this important differentiation! We have changed the paragraph to point to the possible admixture more rigorously (now L73-78). 

This is not to say that I disagree with the argumentation that 1) free-ranging dog populations are worth studying (they are and we should), and 2) that some of them are better representatives than breed dogs. I just think that especially point 2) should be approached with caution and maybe in this context, where the authors do not in fact have any evidence for the genetic makeup of their free-ranging dog population, should err on the side of the narrative of point 1) and simply argue that because the vast majority of the worlds dog population is free-ranging AND thus free of strong artificial selection, we should simply make the effort to develop validated methods to quantify the behaviors of these dogs. That alone is a strong and clear argument – just like in the abstract. The other narrative (i.e. point 2)) is not important to the methods presented and just adds clutter and confusion. 

Answer: We understand your point, and it has merit! We concede that we dwelled on the domestication part too much in the face of what this paper is actually about, particularly in this indicated part. Nevertheless, we do not want to overlook the value free-ranging populations can have for this topic altogether, and we see great importance in advocating for it. And since the paragraph is about the value of testing free-ranging dogs in general rather than our specific population, we argue that our current lack of knowledge about the latter’s genetics does not invalidate that point. This paper is but a starting point rather than an insight into this population in particular. Hence, we retained part of our argumentation here but shortened it and phrased it more adequately to honour the very valid point you made above about admixture interferences (see point above, L73-78).

Line 93-120 is what it is all about – rigor and repeatability, love to see it!

Answer: Now L82-111. We are very grateful to hear that! This part has remained largely unchanged besides your comments below and the writing-related comments made by reviewer 2. 

Specific comments:

Line 112: Interchanging use of free-ranging dogs and street dogs throughout the manuscript, which is a bit confusing.

Answer: We chose to alternate between street and free-ranging dogs on purpose to differentiate between our population and free-ranging dogs overall. While we argue for the importance of considering and testing free-ranging dogs, our sample focussed on street dogs, as defined in L56 and declared in L101. Hence, we mostly use free-ranging in the intro and then switch to street dogs in the methods and results. In the discussion, we alternate depending on whether it pertains to free-ranging populations overall or whether we can only infer to (these) street dogs in particular. While we understand that this might be a bit confusing, we do not want to make overgeneralized statements that we do not have any evidence for, and we trust the readers to understand the difference after rereading the definition in case confusion does occur. Nevertheless, to further aid this understanding, we added the definition again when introducing our methods at the end of the introduction (L101). 

Line 114: Which domestication hypotheses. Why is this relevant to this methods paper?

Answer: These were initially mentioned in the first line of the paper but we have removed them from there now. Hence, we added them in brackets here (now L103). 

Line 186: Seems like the authors are testing multiple behaviors at the same time here. Playfulness could be independent of sociability, no? Will sociability and play be separated in the analyses? Ah, I can see that playfulness was dropped from the analyses…

Answer: Indeed, we were testing sociability behaviours as well as play behaviour in this subtest as indicated in L165, divided by the different phases (Human approach – play). The two behaviour categories would have been analysed separately (as suggested by the different categories in the ethogram), but as you correctly identified, play had to be excluded due to low engagement. 

Line 187: My objections to using the fake dog as a test of conspecific aggression from the first review remain unchanged given the results presented (even though the issues with the test possibly quantifying neophobia and boldness instead of conspecific aggression is now lightly touched upon in line 530-532).

Answer: We do agree that a fake dog is not the most ideal measure to assess conspecific behaviour. Its merits in this situation (safety, time-effectiveness, flexibility, standardisation) and the fact that is has been validated repeatedly, as well as the significant drawbacks (see paragraph on the fake dog (L511-521) and in the limitations (L586-589)) are discussed in the discussion and we make suggestions for better approaches (shorter time, real life observations). Since it is the goal of this paper to present our used methods and discuss their pros and cons in depth, we will retain this part in the methods as is and trust the reader to make their own final judgement depending on their situation. 

Line 211: Please also see Nawroth, C., Wiesmann, K., Schlup, P. et al. Domestication and breeding objective did not shape the interpretation of physical and social cues in goats (Capra hircus). Sci Rep 13, 19098 (2023) https://doi.org/10.1038/s41598-023-46373-9 for the latest on the pointing debate.

Answer: Thank you for this lovely read! It was very insightful. We have added it as a reference towards the point you make right below (L472).

Line 479-483: Not necessarily. These dogs could also have learned that humans mean food, which seems highly likely given that they are all scavengers. So, the behaviors the authors here associate with consistency and heritability could simply be the result of a learned behavioral process. How can the authors disentangle these concepts when they do not know the background (genetic, social etc.) of the dogs they are testing? This part is very speculative at best.

Answer: We agree that this part is speculative, and it was not intended otherwise. We did not mean to argue that it is hereditary in our population but rather point to other papers that have provided such evidence. Regardless, you are undoubtedly correct that we cannot disentangle whether life experience or genetics are the reason for such consistency here. We have added a few sentences in the paragraph (L470-472) to bring out that point and a possible argument that might nevertheless suggest a genetic component. 

Line 502-511: This is a thoughtful paragraph. I wonder if point 3) does not leave room for habituation to the test situation, which might affect behavioral measurements?

Answer: Now L494: That is absolutely correct. We had some further elaborations towards those points initially but deleted it to decrease the word count, hoping that the reader could infer as much themselves. But we do agree that particularly for people not familiar with such testing methods, it should be pointed out. One solution might be that the time before the second testing should be substantially longer than 6 weeks, but seeing as we don't know when they would stop remembering the test, that is not ideal. Likewise, just starting from where the initial test was stopped might alter the behaviour in the subsequent subtests as well. Hence, we now added a half-sentence suggesting that the best way might be to repeat the test (for the sake of the missing subtests) but take the measures from the initial test for the subtests that are being repeated to exclude habituation. However, which approach is ultimately the best depends on the methods, goal, and testing constraints, and should thus be selected by the respective researchers themselves.

Line 565-566: The authors have not introduced the domestication syndrome until this point. The reader does not know that this syndrome, or neoteny, is and it is not relevant to the narrative. Strongly suggest deleting this part.

Answer: Thank you for this observant comment. We did delete the reference to the DS and neoteny and simply referred to retained play behaviour as a frequent trait in domesticated animals (L552). 

Line 571-575: This part is only relevant if the authors know for sure that their dog population predates breed formation and they currently do not have the genetic evidence to back this statement up. In this light I find this line of argumentation irrelevant and speculative in regard to the results presented.

Answer: This part was taken out.

Reviewer #2: 

I thank the Authors for addressing previous comments. The scope is much more constrained and I find it has improved the work significantly, resulting in a methods paper that domestication workers will find useful in the design of larger studies, and very informative on the challenges of behavioral studies outside of laboratory environments.

This manuscript is much improved, has better focus, and most importantly, now covers its intended scope and stays within it. The study offers a behavioral testing method for street or free-roaming dogs, and presents results on the reliability of coding and retesting in a non-laboratory environment. It is presented as the longest and most complex behavioral test for this type of setting and sample. The methods are clearly described and the protocol is well explained. The suite of tests is reported to be successful in the amount of coding reliability, and less so in the retesting consistency, but overall, is informative on the feasibility of this type of behavioral assessment and on the challenges that can be encountered. It was a very interesting read, and I learned much while doing so. I am certain that the study adds great value to domestication studies focused on behavioral differences across populations, and hopefully will serve as a starting point for larger works. Importantly, it points out certain pitfalls in standard protocols that did not work well in a street environment, all which could improve and inform future work in this area. It is very interesting that the most consistent traits were those involving human engagement/communication.

I find the Abstract needs work. The wording is awkward in several sentences, and it makes the setup of the article confusing. There were several problems with sentence structure throughout the MS. Please see the pdf where I tracked the suggested edits, as well as edits to the Abstract. Of course, these are only suggestions. 

General answer to reviewer #2: 

Thank you very much for this detailed effort! We have taken your suggestion seriously and changed the ms accordingly (see below for detail). Since reviewer 1 expressed his great appreciation for the abstract, we chose to keep changes in this part to a minimum – besides honouring your suggestions. We hope this has created an easier read that is satisfactory for readers from many different scopes. 

A few more specific comments + In-text comments: 

Abstract

L24: lagging behind what?

Answer: This sentence has been changed to “has been slow” to take out the need for comparison.

L25: THis is another confusing, awkward sentence. You need to keep the subject of the sentence clear. "no tests have been conducted on [what]..."their [subject here is the tests, not the dogs, but structure the sentence so this is clear] reliability is unknown.."

Answer: Thank you for pointing this out. The sentences have been restructured to provide more clarity (L24-28).

L28; i suggest "on"

Answer: Accepted.

L 40: I am not sure there is evidence of this in your paper. Is there? I agree with that your results highlight difficulties in controlling the environment, but I missed how they find evidence of "intrinsic differences between pet/street dogs.

Answer: We were referring particularly to the play part in the human approach test here where a standardly used play and sociability test did not work at all with the free-ranging population since they did not seem to understand this concept. This intrinsic difference might be due to both learned and potentially genetic differences, as argued in L559-562. 

L40: Reword: "bearing in mind the suggestions made in the discussion". Otherwise, sentence does not make sense. I would even start with""This test battery...." and removing the phrase before.

Answer: Now L41: The “bearing in mind” part has been deleted and replaced with “with some adaptations” since we did not want to advocate for this test battery in its unchanged form but do agree that the sentence structure was awkward.

Intro

L47: should be plural

Answer: Accepted.

L49: (a)if you start a phrase with "from" you need to follow up with "to"..... (b) multiple aspects of [what??] This whole phrase is vague...please specify what you are trying to say.

Answer: In order to shorten the intro as per reviewer 1’s request, these two sentences were merged, which in turn deleted the words you commented on here. Nevertheless, we thank you for this useful suggestion. 

L51: Again, awkward sentence structure...perhaps "there has been a surge in comparative studies..?""

Answer: Same as above.

L54: I suggest either "Importantly"or "however", but not both.

Answer: Thank you. We chose to stick with “however”.

L57: Wow. Very interesting.

Answer: Right?! Hence the need to include them more.

L63: here you imply this is compared to something...compared to [what?]

Answer: Useful observation, thank you. For the sake or decreasing wordiness, we simply deleted this word. 

L84: You would need to specify what "this" is, here.

Answer: As above, this part has been rewritten for the sake of wordiness and better nuancing. 

L87; should be plural

Answer: Correct, thank you. However, this was deleted in line with reviewer 1’s comment about nuancing this paragraph.

L93: I suggest "on"

Answer: Accepted.

L95: "in which they reside". I strongly encourage never ending a sentence with a preposition.

Answer: A valuable suggestion, thank you. Accepted. 

L103: rater's.

Answer: Accepted.

L111: Grammatical suggestion: either (a) you "address" an issue or problem... (b) or you "have" objectives.

Answer: We have changed this to “achieve an objective”.

L120: "to be used more.."

Answer: Since we do not only refer to a higher frequency but also a more widespread use, we changed this phrasing to "more widely used".

Methods

L202: this is not the definition of neophobia, correct? I suggest correcting this, it is confusing. // Did the authors err in their definition of “neophobia”? Or is it perhaps that in the writing style it seems as if they misinterpret its meaning? Please check. (Methods section).

Answer: Thank you for pointing this out. It was indeed not meant as the definition but rather as a clarification for all the assessable target behaviours. The line has been written more clearly (L193).

Discussion

L485: this sentence is unclear...you have differences between 3 things here...did you mean between 2 things?

Answer: This line was indeed confusing, thank you for catching that awkward phrasing. We deleted the obsolete part (L474). 

L499: do not

Answer: Since we referred to the begging test here, the grammar was correct. However, we see how the sentence can be confusing, so we added this part in brackets to refer to the begging test specifically rather than have it as a half sentence (L486). 

L555: 1. Line 555 (Discussion): I am not sure how one could justify making the statement that “behavioral plasticity is selected against in modern [dog] breeds. But if made, please provide a citation or source. // Please provide a citation or supportive evidence for this from a source. I am not sure how one could justify making this statement that behavioral plasticity is selected against in modern breeds. 

Answer: We do not mean to suggest that it was consciously selected against but rather that it might have been decreased through the selection for stable behavioural traits and the loss of genetic variation associated with that. This has been rewritten accordingly to make that point clearer and references have been added for the latter two points (L540-542). 

L624: Reword to: "Bearing in mind the suggestions made..."

Answer: Accepted.

---

## [Decision Letter · Decision Letter 2]

4 Dec 2023

PONE-D-23-22870R2Street-wise dog testing: feasibility and reliability of a behavioural test battery for free-ranging dogs in their natural habitatPLOS ONE

Dear Dr. Capitain, Thank you for submitting your manuscript to PLOS ONE. After careful consideration, we feel that it has merit but does not fully meet PLOS ONE’s publication criteria as it currently stands. Therefore, we invite you to submit a revised version of the manuscript that addresses the points raised during the review. Please submit your revised manuscript by Jan 18 2024 11:59PM. If you will need more time than this to complete your revisions, please reply to this message or contact the journal office at plosone@plos.org. Please include the following items when submitting your revised manuscript:A rebuttal letter that responds to each point raised by the academic editor and reviewer(s). You should upload this letter as a separate file labeled 'Response to Reviewers'.A marked-up copy of your manuscript that highlights changes made to the original version. You should upload this as a separate file labeled 'Revised Manuscript with Track Changes'.An unmarked version of your revised paper without tracked changes. You should upload this as a separate file labeled 'Manuscript'.If applicable, we recommend that you deposit your laboratory protocols in protocols.io to enhance the reproducibility of your results. Protocols.io assigns your protocol its own identifier (DOI) so that it can be cited independently in the future. For instructions see: https://journals.plos.org/plosone/s/submission-guidelines#loc-laboratory-protocols. Additionally, PLOS ONE offers an option for publishing peer-reviewed Lab Protocol articles, which describe protocols hosted on protocols.io. Read more information on sharing protocols at https://plos.org/protocols?utm_medium=editorial-email&utm_source=authorletters&utm_campaign=protocols.

We look forward to receiving your revised manuscript.

Kind regards,

Joshua Kamani, PhD

Academic Editor

PLOS ONE

Journal Requirements:

Additional Editor Comments:

Dear authors

I am glad to inform you that your manuscript have been reviewed and you need to address some minor issues.

Reviewers' comments:

Reviewer's Responses to Questions

**Comments to the Author**

1. If the authors have adequately addressed your comments raised in a previous round of review and you feel that this manuscript is now acceptable for publication, you may indicate that here to bypass the “Comments to the Author” section, enter your conflict of interest statement in the “Confidential to Editor” section, and submit your "Accept" recommendation.

Reviewer #1: All comments have been addressed

Reviewer #2: (No Response)

2. Is the manuscript technically sound, and do the data support the conclusions?

Reviewer #1: Yes

Reviewer #2: Yes

3. Has the statistical analysis been performed appropriately and rigorously? 

Reviewer #1: Yes

Reviewer #2: I Don't Know

4. Have the authors made all data underlying the findings in their manuscript fully available?

Reviewer #1: Yes

Reviewer #2: Yes

5. Is the manuscript presented in an intelligible fashion and written in standard English?

Reviewer #1: Yes

Reviewer #2: Yes

6. Review Comments to the Author

Reviewer #1: I have no further comments to this manuscript.

Congratulations on a well-revised submission and Happy Holidays!

Reviewer #2: Dear Authors,

Thank for your for addressing my edits. I hope they have proved useful.

I enjoyed learning about this subfield from your manuscript.

The manuscript is clearer, has better direction and accomplishes its goals of providing a baseline for behavioral tests of street/free-ranging dogs. It was very interesting to see how these tests are conducted, what the challenges are, and that they can, in fact, provide reliable information about certain behaviors in the variable conditions outside the “pet” environment.

7. PLOS authors have the option to publish the peer review history of their article (what does this mean?). If published, this will include your full peer review and any attached files.

Reviewer #1: No

Reviewer #2: No

---

## [Author Response · Author response to Decision Letter 2]

6 Dec 2023

EDITOR 2:

I suggest some minor edits that will simply clarify that this is not a domestication study, but a behavioral study in the street environment. It will be informative to domestication workers to compare results from this model, but it is not a domestication itself. My apologies if this was not clear before, or if I caused any confusion.

The edits should be quick.

I am happy to clarify anything, if needed, I sometimes write quickly (!).

GENERAL ANSWER: 

Thank you very much for the kind words and the clarifications. Since it seems to be quite controverse and easy to misunderstand, we decided to take out the domestication part from the conclusions. The suggested edits have been made in the rest of the manuscript and the question about the citation has been answered below. We greatly appreciate your diligent work and your constructive suggestions along this paper’s journey! 

REVIEWER 2 DETAILS :

Lines76-78. Please see my comment on distinguishing between this study and domestication research:

In order not to be misleading, I suggest changing to something akin to:

"…this study provides insights for comparison with domestication research…"

This is *not* a domestication study.

I repeat that it will add value to domestication research by providing a comparative model from street populations, but it is not an insight into domestication work itself.

Answer: As described above, we decided to take out the point about domestication from the conclusions entirely in order to not mislead the reader or add to many words for clarification. 

Line 428. I think you did not mean this "and" here? PLease ignore if I am wrong

Answer: A very good catch, thank you very much! The “and” has been deleted

Line 505: YOu may wish to correct the run-on structure here.

Answer: Thank you, this has been changed for a better reading experience.

Line 539. This is rather an important conclusion/position. I wonder if there is a citation here, or if the authors are the first to find this?

Answer: This is solely a speculative suggestion we make based on our observations, hence the lack of citation. We made this a bit clearer in the text now. So far, there is very little research into behavioural flexibility in dogs (to our knowledge) apart from the meta-analysis in breed dogs cited below. Further research testing this idea would therefore be a very interesting avenue.

Line 546: question "of" whether...

Answer: Changed accordingly

Line 599. Conclusions. Again, I think the study is valuable because one can compare a street model vs a domestication model, but is not suitable for "testing questions about domestication" itself. This distinction must be clear, and should not take away from the value of the study.

Answer: Now Line 602: We agree that this was phrase a bit too undifferentiated. The suggested specifications have been implemented.

---

## [Editor Report · Decision Letter 3]

15 Dec 2023

Street-wise dog testing: feasibility and reliability of a behavioural test battery for free-ranging dogs in their natural habitat

PONE-D-23-22870R3

Dear Dr. Svenja Capitain,

We’re pleased to inform you that your manuscript has been judged scientifically suitable for publication and will be formally accepted for publication once it meets all outstanding technical requirements.

Kind regards,

Joshua Kamani, PhD

Academic Editor

PLOS ONE
---

## [Editor Report · Acceptance letter]

21 Dec 2023

PONE-D-23-22870R3 

PLOS ONE

Dear Dr. Capitain, 

I'm pleased to inform you that your manuscript has been deemed suitable for publication in PLOS ONE. Congratulations! Your manuscript is now being handed over to our production team.

Kind regards, 

on behalf of

Dr. Joshua Kamani 

Academic Editor

PLOS ONE